# ON THE CONVERGENCE OF NONCONVEX CONTINUAL LEARNING WITH ADAPTIVE LEARNING RATES

## ABSTRACT

One of the objectives of continual learning is to prevent catastrophic forgetting in learning multiple tasks sequentially. The memory based continual learning stores a small subset of the data for previous tasks and applies various methods such as quadratic programming and sample selection. Some memory-based approaches are formulated as a constrained optimization problem and rephrase constraints on the objective for memory as the inequalities on gradients. However, there have been little theoretical results on the convergence of continual learning. In this paper, we propose a theoretical convergence analysis of memory-based continual learning with stochastic gradient descent. The proposed method called nonconvex continual learning (NCCL) adapts the learning rates of both previous and current tasks with the gradients. The proposed method can achieve the same convergence rate as the SGD method for a single task when the catastrophic forgetting term which we define in the paper is suppressed at each iteration. It is also shown that memory-based approaches inherently overfit to memory, which degrades the performance on previously learned tasks. Experiments show that the proposed algorithm improves the performance of continual learning over existing methods for several image classification tasks.

## 1 INTRODUCTION

Learning new tasks without forgetting previously learned tasks is a key aspect of artificial intelligence to be as versatile as humans. Unlike the conventional deep learning that observes tasks from an i.i.d. distribution, continual learning train sequentially a model on a non-stationary stream of data (Ring, 1995; Thrun, 1994). The continual learning AI systems struggle with catastrophic forgetting when the data access of previously learned tasks is restricted (French & Chater, 2002). To overcome catastrophic forgetting, continual learning algorithms introduce novel methods such as a replay memory to store and replay the previously learned examples (Lopez-Paz & Ranzato, 2017; Aljundi et al., 2019; Chaudhry et al., 2019a), regularization methods that penalize neural networks (Kirkpatrick et al., 2017; Zenke et al., 2017), Bayesian methods that utilize the uncertainty of parameters or data points (Nguyen et al., 2018; Ebrahimi et al., 2020), and other recent approaches (Yoon et al., 2018; Lee et al., 2019). In this paper, we focus on the online continual learning with replay memory. The learner stores a small subset of the data for previous tasks into a memory and utilizes the memory by replaying samples to keep the model staying in a feasible region corresponding to moderate suboptimal region. Gradient episodic memory (GEM) (Lopez-Paz & Ranzato, 2017) first formulated the replay based continual learning as a constrained optimization problem. This formulation rephrases the constraints on objectives for previous tasks as the inequalities based on the inner product of loss gradient vectors for previous tasks and a current task. However, the theoretical convergence analysis of the performance of previously learned tasks, which implies a measure of catastrophic forgetting, has not been rigorously studied in the literature. Without convergence analysis, this intuitive reformulation of constrained optimization does not provide theoretical guarantee to prevent catastrophic forgetting.

Nonconvex finite-sum optimization problem offers a solution to analyze catastrophic forgetting by measuring the convergence of previously learned tasks, which is related to the performance. Most deep learning problems are defined as nonconvex optimization, and the target objective is composed of the sum of objectives for each data point. Now we express our continual learning problem of the

form

$$\min_{x \in \mathbb{R}^d} f(x) = \frac{1}{n} \sum_{i=1}^{n} f_i(x), \tag{1}$$

where we assume that each objective $f_i(x)$ with a model $x$ and a data point $i$ is nonconvex with Lipschitz gradient. Here, we expect that a stochastic gradient descent based algorithm reaches a stationary point instead of the global minimum in the nonconvex optimization. This generic form is well studied to demonstrate the convergence and complexity of stochastic gradient methods for a nonconvex setting (Zhou & Gu, 2019; Lei et al., 2017; Reddi et al., 2016a; Zaheer et al., 2018). Unlike the convex case, the convergence is generally measured by the expectation of the squared norm of the gradient $\mathbb{E}\|\nabla f(x)\|^2$. The theoretical complexity is derived from the $\epsilon$-accurate solution, which is also known as a stationary point with $\mathbb{E}\|\nabla f(x)\|^2 \leq \epsilon$. Suppose we divide the entire sum of objectives into two terms for previous tasks and current tasks, and measure the convergence on each term. Then, we can observe the transition of convergences on the previous and current tasks respectively while learning sequentially from a data stream. We consider this transition of convergence on the previous task as catastrophic forgetting if $\mathbb{E}\|\nabla f_P(x)\|^2$ with the set of data points from previous tasks $P$ increases over iterations.

In this work, we formulate continual learning problem as a nonconvex finite-sum optimization with stochastic gradient descent algorithm that updates both previously learned tasks from replay memory and the current task simultaneously, and present a theoretical convergence analysis of continual learning problems by leveraging the replay-based update method. This extends the continual learning algorithm such as ER-Reservoir (experience replay with reservoir sampling) with fixed and same learning rates on both two tasks to our adaptive method which controls the relative importance between tasks at each step with theoretical guarantee. In addition, the replay-based continual learning has the critical limitation of overfitting to memory, which also degrades the performance of previously learned tasks like catastrophic forgetting by interference. It is known that choosing the perfect memory for continual learning to prevent catastrophic forgetting is an NP-hard problem by (Knoblauch et al., 2020). We present that the inductive bias by replay memory, which prevents perfect continual learning is inevitable in view of optimization.

## 2 BACKGROUNDS

The continual learning algorithm with a replay memory with size $m$ cannot access the whole dataset of the previously learned tasks with $n_f$ samples, but uses limited samples in the memory when a learner trains on the current task. This limited access does not guarantee to completely prevent catastrophic forgetting, and causes the overfitting problem with the biased gradient on a memory. In Section 3, we provide the convergence analysis of the previously learned tasks $f(x)$, which are vulnerable to catastrophic forgetting.

As we denote $f_i(x)$ as the component, which indicates the loss of sample $i$ from the previously learned tasks with the model parameter $x$, we define $\nabla f_i(x)$ as its gradient. We use $I_t$, $J_t$ as the mini-batch of samples at iteration $t$ and denote the mini-batch size $|I_t|$ and $|J_t|$ as $b_f$, $b_g$ throughout the paper. We also note that $g_j(x)$, which denotes the loss for the current task and will be defined in Section 3.1, satisfies the above and following assumptions.

To formulate the convergence over iterations, we introduce the Incremental First-order Oracle (IFO) framework (Ghadimi & Lan, 2013), which is defined as a unit of cost by sampling the pair $(\nabla f_i(x), f_i(x))$. For example, a stochastic gradient descent algorithm requires the cost as much as the batch size $b_t$ at each step, and the total cost is the sum of batch sizes $\sum_{t=1}^{T} b_t$. Let $T(\epsilon)$ be the minimum number of iterations to guarantee $\epsilon$-accurate solutions. Then the average bound of IFO complexity is less than or equal to $\sum_{t=1}^{T(\epsilon)} b_t$.

To analyze the convergence and compute the IFO complexity, we define the supremum of loss gap between a local optimal point $x^0$ and the global optimum $x^*$ as

$$\Delta_f = \sup_{x^0} f(x^0) - f(x^*). \tag{2}$$

Suppose that $\sup f(x^0)$ is same with $f(x^*)$, then we have $\Delta_f = 0$, which might be much smaller than the loss gap of general SGD. Without the continual learning scenario, a general nonconvex SGD

updates the parameters from an randomly initialized point, which is highly likely to have the large loss $f(x^0)$. Then, $\Delta_f > 0$ is the key constant to determine the IFO complexity for convergence as $\Delta_f$ is in the numerator of Equation 11. However, a continual learning algorithm has already converged to a local optimal point $x^0$ for the previous task $f(x)$ and might get a much smaller $\Delta_f$ than the general SGD. It means that $\Delta_f$ for nonconvex continual learining in Equation 11 dose not have a large impact on the IFO complexity. To generalize the theoretical result, we define the worst local minimum to explain the upper bound of convergence rate in Equation 2. This implies that $\Delta_f$ is not a critical reason for moving away from stationary points of $f$ by catastrophic forgetting, which we will explain in Section 3.

We also define $\sigma_f$ and $\sigma_g$ for $f$ and $g$, respectively, as the upper bounds on the variance of the stochastic gradients of a given mini-batch. For brevity, we write only one of them $\sigma_f$,

$$\sigma_f = \sup_x \frac{1}{n_f} \sum_{i=1}^{n_f} \|\nabla f_i(x) - \nabla f(x)\|^2. \tag{3}$$

Throughout the paper, we assume the $L$-smoothness.

**Assumption 1.** *$f_i$ is $L$-smooth that there exists a constant $L > 0$ such that for any $x, y \in \mathbb{R}^d$,*

$$\|\nabla f_i(x) - \nabla f_i(y)\| \leq L \|x - y\| \tag{4}$$

*where $\|\cdot\|$ denotes the Euclidean norm. Then the following inequality directly holds that*

$$-\frac{L}{2}\|x - y\|^2 \leq f_i(x) - f_i(y) - \langle \nabla f_i(y), x - y \rangle \leq \frac{L}{2}\|x - y\|^2. \tag{5}$$

We derive the inequality in Appendix B. With Assumption 1, we can successfully handle individual noncovex objectives for each data point. In the next section, we investigate nonconvex continual learning with adaptive learning rates to overcome catastrophic forgetting.

## 3    NONCONVEX CONTINUAL LEARNING

We first present a theoretical convergence analysis of memory-based continual learning in nonconvex setting. We use the convergence rate of stochastic gradient methods, which denotes the IFO complexity to reach an $\epsilon$-accurate solution for smooth nonconvex finite-sum problem (Reddi et al., 2016a). This generic form enables both deep learning and optimization communities to formulate various accelerated gradient methods with theoretical guarantee. We seek to understand why catastrophic forgetting happens in terms of the convergence rate, and propose non-convex continual learning (NCCL) algorithms with theoretical convergence analysis.

### 3.1    PROBLEM FORMULATION

Given two finite sets $P$ and $C$ at the initial time step $t = 0$, we let two sets denote the sets of indices for previously learned data points and upcoming data points, respectively. Note that the task description for a continual learner is two separate sets. In this section, we will show a convergence analysis of the model parameter that we have trained on $P$ and starts to learn $C$. Thus, we simply denote a data stream of continual learning as two consecutive sets $P$ and $C$.

We consider our goal as a smooth nonconvex finite-sum optimization problem with two objectives

$$\min_{x \in \mathbb{R}^d} F(x) = f(x) + g(x) = \frac{1}{n_f} \sum_{i \in P} f_i(x) + \frac{1}{n_g} \sum_{j \in C} g_j(x), \tag{6}$$

where $f_i(x)$ and $g_j(x)$ denote the objectives of data points $i \in P$ and $j \in C$, respectively. In addition, $n_f$ and $n_g$ are the numbers of elements for $P$ and $C$. To ease exposition, we use a different notation $g_j(x)$ for a data point $j \in C$, which is usually the same objective function for a data point $i \in P$.

To formulate a theoretical convergence analysis of continual learning, we consider a replay memory based method of which memory is a subset of $P \cup C$. Let a random variable $M_t \subset P \cup C$ be the replay memory at time step $t \in [0, T]$, whose union is of the form $M := \cup_t M_t$. We focus both the

episodic memory and the replay memory with dropping rule. The episodic memory based methods include GEM (Aljundi et al., 2019), A-GEM (Chaudhry et al., 2019a), and ORTHOG-SUBSPACE (Chaudhry et al., 2020). ER-Reservoir (Chaudhry et al., 2019b) is a replay memory based method with dropping rule, which replaces the dropped sample $d \in M_t$ with a sample in the stream for $C$.

We now define the gradient update of continual learning

$$x^{t+1} = x^t - \alpha_{H_t} \nabla f_{I_t}(x^t) - \beta_{H_t} \nabla g_{J_t}(x^t), \tag{7}$$

where $I_t \subset M_t$ and $J_t \subset C$ denote the mini-batches from the replay memory and the current data stream, respectively. Here, $H_t$ is the union of $I_t$ and $J_t$. The adaptive learning rates of $\nabla f_{I_t}(x^t)$ and $\nabla g_{J_t}(x^t)$ are denoted by $\alpha_{H_t}$ and $\beta_{H_t}$ which are the functions of $H_t$. Strictly speaking, the mini-batch $I_t$ from $M_t$ might contain a datapoint $d \in C$ for ER-Reservoir. We describe the details of the problem in Appendix B and assume that the notation $I_t$ indicates a subset of $P$ for convenience. Equation 7 is a generalized version of continual learning algorithms, which is our novelty to prove the convergence rates in the nonconvex setting for the proposed method, A-GEM, and ER-Reservoir later.

## 3.2 MEMORY-BASED NONCONVEX CONTINUAL LEARNING

Unlike the conventional smooth nonconvex finite-sum problem where each mini-batch is iid-sampled from the dataset $P \cup C$, the replay memory based continual learning encounters a non-iid stream of data $C$ and has access to a small sized memory $M$. Algorithm 1 presents the pseudocode of NCCL by Equation 7. By the limited access to $P$, the gradient update for $f(x)$ in Equation 7 is a biased estimate of the gradient $\nabla f(x^t)$. Specifically, at the timestep $t$,

$$\nabla f_{M_t}(x^t) = \mathbb{E}_{I_t} \left[ \nabla f_{I_t}(x^t)|M_t \right] = \mathbb{E}_{I_t} \left[ \nabla f(x^t) + e_t|M_t \right] = \nabla f(x^t) + e_{M_t},$$

where $e_t$ and $e_{M_t}$ denote the error term $\nabla f_{I_t}(x^t) - \nabla f(x^t)$ and the expectation value over $I_t$ given $M_t$, respectively. We note that the given replay memory $M_t$ with small size at timestep $t$ induces the inevitable overfitting bias. We first state an intermediate result for a single gradient update of NCCL. For ease of exposition, we define the overfitting term $B_t$ and the catastrophic forgetting term $\Gamma_t$ as follows:

$$B_t = (L\alpha_{H_t}^2 - \alpha_{H_t})\langle \nabla f(x^t), e_t \rangle + \beta_{H_t}\langle \nabla g_{J_t}(x^t), e_t \rangle,$$

$$\Gamma_t = \frac{\beta_{H_t}^2 L}{2} \|\nabla g_{J_t}(x^t)\|^2 - \beta_{H_t}(1 - \alpha_{H_t}L)\langle \nabla f_{I_t}(x^t), \nabla g_{J_t}(x^t) \rangle,$$

where $L$ is a constant for Lipschitz smoothness. Under Assumption 1, a single gradient update by Equation 7 satisfies the following bound by letting $x \leftarrow x^{t+1}$ and $y \leftarrow x^t$:

$$\left(\alpha_{H_t} - \frac{L}{2}\alpha_{H_t}^2\right)\|\nabla f(x^t)\|^2 \leq f(x^t) - f(x^{t+1}) + \Gamma_t + B_t + \frac{L}{2}\alpha_{H_t}^2\|e_t\|^2. \tag{8}$$

This reveals the basic qualitative difference between the conventional nonconvex SGD and NCCL in the convergence rate. Compared to the nonconvex SGD, there exist two terms $B_t$ and $\Gamma_t$ in Equation 8. We group the terms containing $e_t$ as $B_t$ and the other terms as $\Gamma_t$. We note that the catastrophic forgetting term $\Gamma_t$ has $\langle \nabla f_{I_t}(x^t), \nabla g_{J_t}(x^t) \rangle$, which is the key aspect of interference and transer (Riemer et al., 2018), and $B_t$ includes the error term between the batch of $M$ and the entire dataset $P$. Then, we can quantify the amount of overfitting by tracing $B_t$.

To compute the expectation over the stochasticity of NCCL, we derive the expectation of $\nabla f_{M_t}(x^t)$ over the sampling rule of $M_t$. The episodic memory $M_t = M_0$ for all $t$ is uniformly sampled once from the random sequence of $P$, and ER-reservoir iteratively samples the replay memory $M_t$ by the selection probability $P(M_t|M_{t-1})$.

**Lemma 1.** *Let the history of $M_t$ be $M_{[0:t]} = (M_0, \cdots, M_t)$. If $M_0$ is uniformly sampled from $P$, then both episodic memory and ER-reservoir satisfies*

$$\mathbb{E}_{M_{[0:t]}} \left[ \nabla f_{M_t}(x^t) \right] = \nabla f(x^t) \quad and \quad \mathbb{E}_{M_{[0:t]}} [e_{M_t}] = 0. \tag{9}$$

We provide the detailed proof in Appendix B. Note that taking expectation iteratively with respect to the history $M_{[0:t]}$ is needed to compute the expected value of gradients for $M_t$. Since taking the expectation over the stochasticity of NCCL implies the total expectation

$$\mathbb{E} = \mathbb{E}_{M_{[0:t]}} \left[ \mathbb{E}_{I_t} \left[ \mathbb{E}_{J_t} \left[ \, \cdot \, |I_t \right] \right] |M_{[0:t]} \right],$$

---

**Algorithm 1** Nonconvex Continual Learning (NCCL)

---

**Input:** Previous task $P$, $K$ task data stream $\{D_1, \cdots D_K\}$, initial model $x^0$, memory $M_0 \subset P$
**for** $k = 1$ **to** $K$ **do**
    **for** $t = 0$ **to** $T - 1$ **do**
        Uniformly sample a mini-batch $I_t \subset M_t$ with $|I_t| = b_f$
        Uniformly sample a mini-batch $J_t \subset D_k$ with $|J_t| = b_g$
        Compute learning rates $\alpha_{H_t}$ and $\beta_{H_t}$ with $\nabla f_{I_t}(x^t)$ and $\nabla g_{J_t}(x^t)$
        $x^{t+1} \leftarrow x^t - \alpha_{H_t} \nabla f_{I_t}(x^t) - \beta_{H_t} \nabla g_{J_t}(x^t)$
        Store $J_t$ into $M_{t+1}$ by the rule of replay memory scheme
    **end for**
    $x^0 \leftarrow x^{T-1}$
    $P \leftarrow P \cup \bigcup_t J_t$
    $M_0 \leftarrow M_{T-1}$
**end for**

---

we know that $\mathbb{E}[B_t]$ is also 0. We note that considering a random choice of $M_0$ allows us to analyze the convergence on $f(x)$, even if $M_0$ cannot access the all datapoints in $P$ to compute $\|\nabla f(x^t)\|^2$. Our first main result is the following lemma that provides the stepwise change of upper bound.

**Lemma 2.** *Suppose that Assumption 1 holds and $0 < \alpha_{H_t} \leq \frac{2}{L}$. Then for $x^t$ updated by Algorithm 1, we have the following bound*

$$\mathbb{E}\|\nabla f(x^t)\|^2 \leq \mathbb{E}\left[\frac{1}{\alpha_{H_t}(1 - \frac{L}{2}\alpha_{H_t})}\left(f(x^t) - f(x^{t+1}) + B_t + \Gamma_t\right) + \frac{\alpha_{H_t}L}{2(1 - \frac{L}{2}\alpha_{H_t})}\sigma_f^2\right], \quad (10)$$

*where the effect of $B_t$ vanishes by $\mathbb{E}[B_t] = 0$.*

The proof is presented in Appendix B. Surprisingly, taking the expectation over $M_0 \subset P$ in Equation 10 to handle the stochasticity of choosing $M_0$ allows us to analyze the convergence of $f(x)$ in Section 3.3. We also note that the individual trial with a randomly given $M_0$ cannot cancel the effect by $B_t$, although its total expectation over the whole possible trials $\mathbb{E}[B_t]$ is zero. More specifically, the worst case $\sup_{M_0} \|\nabla f(x^t)\|^2$ definitely contains the non-zero value of $B_t$ in the upper bound term. We discuss the more details of the overfitting to memory in Appendix D.

### 3.3 THEORETICAL CONVERGENCE ANALYSIS

We now describe our convergence analysis of the stochastic gradient descent by Equation 7. Without any continual learning scheme, the model $x^t$ eventually converges to $M \cup C$, not $P \cup C$. A local optimal point $x^0$ for $P$ at time step $t = 0$, which we have already trained before the continual learning step starts, loses the performance on $P$ very quickly over time. This phenomenon is called catastrophic forgetting. We also note that the overfitting to replay memory is another reason of degrading performance for each trial when the model trains on samples in $M$ with replacement after learning on $P$ for previous tasks. We obtain the convergence rate of NCCL to a stationary point of $f$ as the following theorem.

**Theorem 1.** *Let $\alpha_{H_t} = \alpha = \frac{c}{\sqrt{T}}$ for some $0 < c \leq \frac{2\sqrt{T}}{L}$ and $t \in \{0, \cdots, T-1\}$. By Lemma 2, the iterates of NCCL satisfy*

$$\min_t \mathbb{E}\|\nabla f(x^t)\|^2 \leq \frac{A}{\sqrt{T}}\left(\frac{1}{c}\left(\Delta_f + \sum_{t=0}^{T-1} \mathbb{E}[\Gamma_t]\right) + \frac{Lc}{2}\sigma_f^2\right), \quad (11)$$

*where $A = 1/(1 - L\alpha/2)$.*

For completeness we present a proof in Appendix A. The loss gap $\Delta_f$ and the variance of gradients $\sigma_f$ are fixed values. We first note that there exists the cumulative sum of $\mathbb{E}[\Gamma_t]$ unlike the convergence rate of SGD. Most of novelty in our analysis lies in dealing with $\sum \mathbb{E}[\Gamma_t]$. The $\sum_{t=0}^{T-1} \mathbb{E}[\Gamma_t]/\sqrt{T}$

is not guaranteed to converge to 0. This fact gives rise to catastrophic forgetting in terms of the nondecreasing upper bound.

Before showing a bound on IFO calls made by NCCL, we present the convergence rate for $g(x)$. To simplify the proof, we assume that learning rates $\alpha_{H_t}, \beta_{H_t}$ are a same fixed value $\beta = c'/\sqrt{T}$. The assumption is reasonable, because we can observe that RHS of Equation 10 is not perturbed drastically by the learning rates with small value $0 < \alpha_{H_t}, \beta_{H_t} \leq 2/L \ll 1$. Then we have the following lemma.

**Lemma 3.** *Suppose that $I_t \cap J_t = \emptyset$, and the datapoints $d \in M \cap P$ use the same objective function $g_d = f_d$. Taking expectation over $I_t \subset M_t$ and $J_t \subset C$, we have*

$$\min_t \mathbb{E}\|\nabla g(x^t)\|^2 \leq \sqrt{\frac{2\Delta_g L}{T}}\sigma_g,$$ (12)

*where $\Delta_g$ and $\sigma_g$ is the version of loss gap and the variance for g on $M \cup C$, respectively. In fact, we note that the convergence rate of g is on $M \cup C$. Despite the convergence rate is not on $C$, it also converges to $C$ trivially.*

Now we can get the upper bound of $\sum \mathbb{E}[\Gamma_t]$ by Lemma 3. Then we have the following main result.

**Corollary 1.** *The IFO complexity of Algorithm 1 to obtain an $\epsilon$-accurate solution is:*

$$\textit{IFO calls} = \begin{cases} O(1/\epsilon^2), & \sum \mathbb{E}[\Gamma_t] = O(1), \\ \infty, & \sum \mathbb{E}[\Gamma_t] = O(T). \end{cases}$$ (13)

The proofs of both Lemma 3 and Corollary 1 are presented in Appendix B. Here, we denote $\infty$ as the impossibility of convergence. This result reveals that catastrophic forgetting is inevitable without minimizing the cumulative sum $\sum \mathbb{E}[\Gamma_t]$. With replay memory, we know that the inner product in $\Gamma_t$ can be negative by the result of A-GEM (Chaudhry et al., 2019a). This observation shows that there exists the chance that $\Gamma_t$ becomes negative, and $\sum \mathbb{E}[\Gamma_t]$ can converge to a constant $O(1)$ as $t \to \infty$. Now we have the proper convergence rate of $f(x)$, and the model can keep the performance on $P$ after learning on $C$. On the other hand, when $\alpha_{H_t} = 0$ for all time step, $\mathbb{E}[\Gamma_t]$ is always larger than 0. Intuitively, this case implies SGD on $C$ without the replay memory $M$ or other supports, and the cumulative sum $\sum \mathbb{E}[\Gamma_t]$ monotonically increases over time. Then, $f(x^t)$ diverges as $t \to \infty$. We emphasize that the reason of catastrophic forgetting can be explained remarkably by the nonconvex optimization perspective.

**The optimization problem of nonconvex continual learning.** The local optimal point $x^t$ after learning on $C$ might be different from $x^0$, because $x^t$ moves towards $x^*_{P \cup C}$ as described in Figure 1. The above observation motivates the following formulation of continual learning to induce $\sum \mathbb{E}[\Gamma_t]$ to converge as $t \to \infty$ and keep $\epsilon$-accuracy of $f$ during $T$ iterations. More specifically, we solve the following problem over $T$:

$$\underset{\alpha_{H_t}, \beta_{H_t}}{\text{minimize}} \quad \sum_{t=0}^{T-1} \Gamma_t$$

$$\text{subject to} \quad 0 < \alpha_{H_t}, \beta_{H_t} \leq 2/L \text{ for all } t < T.$$ (14)

Recall that we have the condition of two learning rates to prove Theorem 1 and Lemma 3. This condition needs to be constraints of the above optimization problem.

Finally, we note that our result first shows a theoretical convergence analysis of memory based nonconvex continual learning in the setting of Corollary 1. More precisely, we can now explain this method by Problem 14 with theoretical guarantee.

## 3.4 ADAPTIVE LEARNING RATES

As discussed in Section 3.3, the minimization problem 14 leads to tight the upper bound of convergence rate in nonconvex setting. The result of convergence analysis provides a simple continual learning framework that only adjusts two learning rates in Equation 7. First, we review A-GEM and other methods with fixed learning rate such as GSS (Aljundi et al., 2019) and ER-Reservoir (Chaudhry et al., 2019b) through our result.

**A-GEM** (Chaudhry et al., 2019a) propose a surrogate of $\nabla g_{J_t}(x^t)$ as the following equation to avoid violating the constraint when $\langle \nabla f_{I_t}(x^t), \nabla g_{J_t}(x^t) \rangle \leq 0$:

$$\nabla g_{J_t}(x^t) - \left\langle \frac{\nabla f_{I_t}(x^t)}{\|\nabla f_{I_t}(x^t)\|}, \nabla g_{J_t}(x^t) \right\rangle \frac{\nabla f_{I_t}(x^t)}{\|\nabla f_{I_t}(x^t)\|}.$$

We can interpret the surrogate as boosting the learning rate $\alpha_{H_t}$ to cancel out the negative component of $\nabla f_{I_t}(x^t)$ on $\nabla g_{J_t}(x^t)$. After applying the surrogate, $\mathbb{E}[\Gamma_t]$ is reduced but still non-negative, which we show in Appendix C. A-GEM also fails to fully use the advantage of the case $\langle \nabla f_{I_t}(x^t), \nabla g_{J_t}(x^t) \rangle > 0$.

**Fixed learning rates** imply that building a replay memory is the key to success in continual learning. It means that this memory setting is an abundant pool to sample $I_t$, which satisfies $\langle \nabla f_{I_t}(x^t), \nabla g_{J_t}(x^t) \rangle > 0$, as much as possible. This can reduce $\sum \mathbb{E}[\Gamma_t]$.

However, two above methods cannot enjoy the case $\langle \nabla f_{I_t}(x^t), \nabla g_{J_t}(x^t) \rangle > 0$ enough. As discussed above, we note that $\Gamma_t$ is a quadratic polynomial of $\beta_{H_t}$ where $\beta_{H_t} > 0$. We can solve the minimum of the polynomial on $\beta_{H_t}$ when $\langle \nabla f_{I_t}(x^t), \nabla g_{J_t}(x^t) \rangle > 0$. By differentiating on $\beta_{H_t}$, we can easily find the minimum $\Gamma_t^*$ and the optimal learning rate $\beta_{H_t}^*$

$$\beta_{H_t}^* = \frac{(1 - \alpha_{H_t} L)\langle \nabla f_{I_t}(x^t), \nabla g_{J_t}(x^t) \rangle}{L\|\nabla g_{J_t}(x^t)\|^2}, \quad \Gamma_t^* = -\frac{(1 - \alpha_{H_t} L)\langle \nabla f_{I_t}(x^t), \nabla g_{J_t}(x^t) \rangle}{2L\|\nabla g_{J_t}(x^t)\|^2}.$$

We integrate the above result to propose a better nonconvex continual learning algorithm with the theoretical guarantee. More specifically, we propose an adaptive learning rate method that can reduce $\sum \mathbb{E}[\Gamma_t]$ in both cases of $\langle \nabla f_{I_t}(x^t), \nabla g_{J_t}(x^t) \rangle \leq 0$ and $\langle \nabla f_{I_t}(x^t), \nabla g_{J_t}(x^t) \rangle > 0$. Then, the proposed adaptive learning rate scheme is as follows.

$$\alpha_{H_t} = \begin{cases} \alpha(1 - \frac{\langle \nabla f_{I_t}(x^t), \nabla g_{J_t}(x^t) \rangle}{\|\nabla f_{I_t}(x^t)\|^2}), & \langle \nabla f_{I_t}(x^t), \nabla g_{J_t}(x^t) \rangle \leq 0 \\ \alpha, & \langle \nabla f_{I_t}(x^t), \nabla g_{J_t}(x^t) \rangle > 0 \end{cases} \tag{15}$$

$$\beta_{H_t} = \begin{cases} \alpha, & \langle \nabla f_{I_t}(x^t), \nabla g_{J_t}(x^t) \rangle \leq 0 \\ \frac{(1 - \alpha L)\langle \nabla f_{I_t}(x^t), \nabla g_{J_t}(x^t) \rangle}{L\|\nabla g_{J_t}(x^t)\|^2}, & \langle \nabla f_{I_t}(x^t), \nabla g_{J_t}(x^t) \rangle > 0 \end{cases} \tag{16}$$

We derive the details of the above result in Appendix C. In addition, our result shows that A-GEM can be viewed as an adaptive learning rate scheme using samples in $M_t$ directly. This fact implies that there is no difference between A-GEM and experience replay methods. Figure 1 illustrates intuitively how scaling learning rates achieve the convergence to a mutual stationary point $x_{P \cup C}^*$ as we proved the theoretical complexity in Corollary 1.

## 4 EXPERIMENTS

We evaluate the proposed NCCL and the state of the art online continual learning baselines on the following benchmarks. We report detailed experimental materials and results in Appendix.

**Datasets.** We demonstrate the experimental results on standard continual learning benckmarks: **Permuted-MNIST** (Kirkpatrick et al., 2017) is a MNIST (LeCun et al., 1998) based dataset, where each task has a fixed permutation of pixels and transform data points by the permutation to make each task distribution unrelated. **Split-MNIST** (Zenke et al., 2017) splits MNIST dataset into five tasks. Each task consists of two classes, for example (1, 7), (3, 4), and has approximately 12K images. **Split-CIFAR10 and 100** also split CIFAR-10 and 100 datasets (Krizhevsky et al., 2009) into five tasks and 20 tasks, respectively.

**Baselines.** We report the experimental evaluation on the online continual setting which implies a model is trained with a single epoch. We compare with the following continual learning baselines. **Fine-tune** is a simple method that a model trains observed data naively without any support, such as replay memory. **Elastic weight consolidation (EWC)** is a regularization based method by Fisher Information (Kirkpatrick et al., 2017). **ER-Reservoir** chooses samples to store from a data stream with a probability proportional to the number of observed data points. The replay memory returns

a random subset of samples at each iteration for experience replay. ER-Reservoir (Chaudhry et al., 2019b) shows a powerful performance in continual learning scenario. **GEM and A-GEM** (Lopez-Paz & Ranzato, 2017; Chaudhry et al., 2019a) use gradient episodic memory to overcome catastrophic forgetting. The key idea of GEM is gradient projection with quadratic programming and A-GEM simplifies this procedure. We also compare with iCarl, MER, ORTHOG-SUBSPACE (Chaudhry et al., 2020), and stable-SGD (Mirzadeh et al., 2020).

**Architecture and Training detail.** For a fair comparison, we follow the commonly used model architecture and hyperparameters of Lee et al. (2020); Chaudhry et al. (2020). For Permuted-MNIST and Split-MNIST, we use fully-connected neural networks with two hidden layers of $[400, 400]$ or $[256, 256]$ and ReLU activation. ResNet-18 with the number of filters $n_f = 64, 20$ (He et al., 2016) is applied for Split CIFAR-10 and 100. All experiments conduct a single-pass over the data stream. It is also called 1 epoch or 0.2 epoch (in the case of split tasks). We deal both cases with and without the task identifiers in the results of split-tasks to compare fairly with baselines. Batch sizes of data stream and memory are both 10. All reported values are the average values of 5 runs with diffrent seeds, and we also provide standard deviation. Other miscellaneous settings are the same as in Chaudhry et al. (2020). We implement the baselines and the propose method on Tensorflow 1. For evaluation, we use an NVIDIA 2080ti GPU along with 3.60 GHz Intel i9-9900K CPU and 64 GB RAM.

## 4.1 RESULTS

The following tables show our main experimental result, which is averaged over 5 runs. We denote the number of example per class per task at the top of each column.

Table 1: Multi-headed split-CIFAR100, reduced size Resnet-18 $n_f = 20$. Accuracy and forgetting results.

| Method | memory size | 1 | | 5 | |
|---|---|---|---|---|---|
| | **memory** | accuracy | forgetting | accuracy | forgetting |
| EWC | ✗ | 42.7 (1.89) | 0.28 (0.03) | 42.7 (1.89) | 0.28 (0.03) |
| Fintune | ✗ | 40.4 (2.83) | 0.31 (0.02) | 40.4 (2.83) | 0.31 (0.02) |
| Stable SGD | ✗ | 59.9 (1.81) | 0.08 (0.01) | 59.9 (1.81) | 0.08 (0.01) |
| A-GEM | ✓ | 50.7 (2.32) | 0.19 (0.04) | 59.9 (2.64) | 0.10 (0.02) |
| ER-Ring | ✓ | 56.2 (1.93) | 0.13 (0.01) | 62.6 (1.77) | 0.08 (0.02) |
| ER-Reservoir | ✓ | 46.9 (0.76) | 0.21 (0.03) | 65.5 (1.99) | 0.09 (0.02) |
| ORTHOG-subspace | ✓ | 58.81 (1.88) | 0.12 (0.02) | 64.38 (0.95) | 0.055 (0.007) |
| NCCL + Ring | ✓ | 54.63 (0.65) | 0.059 (0.01) | 61.09 (1.47) | 0.02 (0.01) |
| NCCL + Reservoir | ✓ | 52.18 (0.48) | 0.118 (0.01) | 63.68 (0.18) | 0.028 (0.009) |

Table 2: Permuted-MNIST (23 tasks 60000 examples per task), FC-[256,256]. Accuracy and forgetting results.

| Method | memory size | 1 | | 5 | |
|---|---|---|---|---|---|
| | **memory** | accuracy | forgetting | accuracy | forgetting |
| multi-task | ✗ | 83 | - | 83 | - |
| Fine-tune | ✗ | 53.5 (1.46) | 0.29 (0.01) | 47.9 | 0.29 (0.01) |
| EWC | ✗ | 63.1 (1.40) | 0.18 (0.01) | 63.1 (1.40) | 0.18 (0.01) |
| MER | ✓ | 69.9 (0.40) | 0.14 (0.01) | 78.3 (0.19) | 0.06 (0.01) |
| A-GEM | ✓ | 62.1 (1.39) | 0.21 (0.01) | 64.1 (0.74) | 0.19 (0.01) |
| ER-Ring | ✓ | 70.2 (0.56) | 0.12 (0.01) | 75.8 (0.24) | 0.07 (0.01) |
| ER-Reservoir | ✓ | 68.9 (0.89) | 0.15 (0.01) | 76.2 (0.38) | 0.07 (0.01) |
| ORHOG-subspace | ✓ | 84.32 (1.10) | 0.12 (0.01) | 84.32 (1.1) | 0.11 (0.01) |
| NCCL + Ring | ✓ | 74.22 (0.75) | 0.13 (0.007) | 84.41 (0.32) | 0.053 (0.002) |
| NCCL+Reservoir | ✓ | 79.36 (0.73) | 0.12 (0.007) | 88.22 (0.26) | 0.028 (0.003) |

Overall, NCCL variants outperforms baseline methods especially in the forgetting metric. Our goal is to demonstrate the usefulness of the adaptive learning rate scheme to reduce the catastrophic forgetting, and verify the proposed theoretical convergence analysis. We remark that our adaptive learning rates successfully suppress forgetting by a large margin compared to baselines. Note that NCCL also outperform A-GEM, which does not maximize transfer $\langle \nabla f_{I_t}(x^t), \nabla g_{J_t}(x^t) \rangle > 0$. Now, we can empirically demonstrate our theoretical guaranteed method by minimizing $\sum \Gamma_t$ is valid.

We clipped $\beta_{H_t}$ to increase the performance. As we discussed earlier, we can prevent forgetting when $\langle \nabla f_{I_t}(x^t), \nabla g_{J_t}(x^t) \rangle > 0$. However, we observe that $\|\nabla f_{I_t}(x^t)\|^2$ suddenly increases because of the interference at the previous step $t - 1$. The very large learning rate $\beta_{H_t}$ by the increased $\|\nabla f_{I_t}(x^t)\|$ can force the model to fall into an arbitrary point that is likely to increase the loss of $f$. Clipping the learning rate reduces this problem and still has the effect of reducing the catastrophic forgetting term $\Gamma_t$. By the property of quadratic polynomial, the catastrophic forgetting term is negative because the clipped value is smaller than the original learning rate.

We show that NCCL is a potentially powerful alternative for continual learning. Even with tiny replay memory, NCCL still performs better than some baselines. We note that NCCL shows the best performance on the forgetting metric. It implies that NCCL prevent catastrophic forgetting more efficiently than others by minimizing the catastrophic forgetting term in the proposed optimization problem. However, the accuracy is slightly lower than other baselines, which include experience replays. The purpose of our adaptive learning rate scheme is to prevent catastrophic forgetting, so the performance of current task is slightly lower than ER-Ring, stable-SGD, and ORTHOG-subspace. This result shows that the plasticity to learn a new task is restricted by NCCL variants with tiny memory. In particular, we would expect that NCCL would benefit from the additional enhancements in ORTHOG-SUBSPACE and stable SGD by introducing their techniques. In appendix, we add more results with larger sizes of memory, which shows that NCCL outperforms on the average accuracy. We conclude that the transfer effect by the small size of memory for NCCL is less effective.

## 5 RELATED WORK

We extend the memory based continual learning (Lopez-Paz & Ranzato, 2017; Chaudhry et al., 2019a; Riemer et al., 2018; Chaudhry et al., 2019b) to the nonconvex optimization problem to provide the theoretical guarantee of previous methods. Our problem setting is related to the theoretical convergence analysis of smooth nonconvex optimization. Smooth nonconvex finite-sum optimization problem has been widely employed to derive the theoretical complexity of computation for stochastic gradient methods (Ghadimi & Lan, 2013; 2016; Lei et al., 2017; Zaheer et al., 2018; Reddi et al., 2016a). Unlike the convex optimization, the gradient based algorithms are not expected to converge to the global minimum but are evaluated by measuring the convergence rate to the stationary points in the nonconvex case. Prior works have only focused on analyze the problems with two objectives where the additional term is for regularization (Yao & Kwok, 2016). Here, we instead focuses on the continual learning problem, which has two objectives for both datapoints of the previous task and the current task. This setting is interesting since we can observe the convergence of each task respectively. Otherwise, (Knoblauch et al., 2020) first develops a theoretical approach on continual learning by set theory, and shows that optimal continual learning algorithms and building a perfect memory is equivalent. This is analogous to our result that we cannot handle the overfitting bias for each trial of continual learning.

## 6 CONCLUSION

In this paper, we have presented the first generic theoretical convergence analysis of continual learning. Our proof shows that a training model can circumvent catastrophic forgetting by suppressing the disturbance term on the convergence of previously learned tasks. We also demonstrate theoretically and empirically that the performance of past tasks by nonconvex continual learning with replay memory is degraded by two separate reasons, catastrophic forgetting and overfitting to memory. To tackle these problems, nonconvex continual learning uses two methods, scaling learning rates adaptive to mini-batches and sampling mini-batches with small size from the replay memory. Finally, it is expected the proposed nonconvex framework is helpful to analyze the convergence rate of other continual learning algorithms.

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

APPENDIX

# A    ADDITIONAL EXPERIMENTAL RESULTS AND DETAILS

## A.1    HYPERPARAMETER GRIDS

We report the hyper-paramters grid we used in our experiments below. Except for the proposed algorithm, we adopted the hyper-paramters that are reported in the original papers. We used grid search to find the optimal parameters for each model.

- finetune - learning rate [0.003, 0.01, 0.03 (CIFAR), 0.1 (MNIST), 0.3, 1.0]
- EWC - learning rate: [0.003, 0.01, 0.03 (CIFAR), 0.1 (MNIST), 0.3, 1.0] - regularization: [0.1, 1, 10 (MNIST,CIFAR), 100, 1000]
- AGEM - learning rate: [0.003, 0.01, 0.03 (CIFAR), 0.1 (MNIST), 0.3, 1.0]
- ER-Reservoir - learning rate: [0.003, 0.01, 0.03 (CIFAR), 0.1 (MNIST), 0.3, 1.0]
- ORTHOG-SUBSPACE - learning rate: [0.003, 0.01, 0.03, 0.1 (MNIST), 0.2, 0.4 (CIFAR), 1.0]
- MER - learning rate: [0.003, 0.01, 0.03 (MNIST, CIFAR), 0.1, 0.3, 1.0] - within batch meta-learning rate: [0.01, 0.03, 0.1 (MNIST, CIFAR), 0.3, 1.0] - current batch learning rate multiplier: [1, 2, 5 (CIFAR), 10 (MNIST)]
- iid-offline and iid-online - learning rate [0.003, 0.01, 0.03 (CIFAR), 0.1 (MNIST), 0.3, 1.0]
- ER-reservoir - learning rate: [0.003, 0.01, 0.03, 0.1 (MNIST, CIFAR), 0.3, 1.0]
- NCCL-Ring (default) - learning rate $\alpha$: [0.003, 0.001(CIFAR), 0.01, 0.03, 0.1, 0.3, 1.0]
- NCCL-Reservoir - learning rate $\alpha$: [0.003(CIFAR), 0.001, 0.01, 0.03, 0.1, 0.3, 1.0]

## A.2    ADDITIONAL RESULTS

Table 3: Multi-headed split-CIFAR100, full size Resnet-18 $n_f = 64$. Accuracy and forgetting results.

| Method | memory size | 1 | | 5 | |
| :---: | :---: | :---: | :---: | :---: | :---: |
| | **memory** | accuracy | forgetting | accuracy | forgetting |
| Fintune | ✗ | 42.6 (2.72) | 0.27 (0.02) | 42.6 (2.72) | 0.27 (0.02) |
| EWC | ✗ | 43.2 (2.77) | 0.26 (0.02) | 43.2 (2.77) | 0.26 (0.02) |
| ICRAL | ✓ | 46.4 (1.21) | 0.16 (0.01) | - | - |
| A-GEM | ✓ | 51.3 (3.49) | 0.18 (0.03) | 60.9 (2.5) | 0.11 (0.01) |
| MER | ✓ | 49.7 (2.97) | 0.19 (0.03) | - | - |
| ER-Ring | ✓ | 59.6 (1.19) | 0.14 (0.01) | 67.2 (1.72) | 0.06 (0.01) |
| ER-Reservoir | ✓ | 51.5 (2.15) | 0.14 (0.09) | 62.68 (0.91) | 0.06 (0.01) |
| ORTHOG-subspace | ✓ | 64.3 (0.59) | 0.07 (0.01) | 67.3 (0.98) | 0.05 (0.01) |
| NCCL + Ring | ✓ | 59.06 (1.02) | 0.03 (0.02) | 66.58 (0.12) | 0.004 (0.003) |
| NCCL + Reservoir | ✓ | 54.7 (0.91) | 0.083 (0.01) | 66.37 (0.19) | 0.004 (0.001) |

Table 4: permuted-MNIST (23 tasks 10000 examples per task), FC-[256,256]. Accuracy and forgetting results.

| Method | memory size | 1 | | 5 | |
|---|---|---|---|---|---|
| | memory | accuracy | forgetting | accuracy | forgetting |
| multi-task | ✗ | 91.3 | - | 83 | - |
| Fine-tune | ✗ | 50.6 (2.57) | 0.29 (0.01) | 47.9 | 0.29 (0.01) |
| EWC | ✗ | 68.4 (0.76) | 0.18 (0.01) | 63.1 (1.40) | 0.18 (0.01) |
| MER | ✓ | 78.6 (0.84) | 0.15 (0.01) | 88.34 (0.26) | 0.049 (0.003) |
| A-GEM | ✓ | 78.3 (0.42) | 0.21 (0.01) | 64.1 (0.74) | 0.19 (0.01) |
| ER-Ring | ✓ | 79.5 (0.31) | 0.12 (0.01) | 75.8 (0.24) | 0.07 (0.01) |
| ER-Reservoir | ✓ | 68.9 (0.89) | 0.15 (0.01) | 76.2 (0.38) | 0.07 (0.01) |
| ORHOG-subspace | ✓ | 86.6 (0.91) | 0.04 (0.01) | 87.04 (0.43) | 0.04 (0.003) |
| NCCL + Ring | ✓ | 74.38 (0.89) | 0.05 (0.009) | 83.76 (0.21) | 0.014 (0.001) |
| NCCL+Reservoir | ✓ | 76.48 (0.29) | 0.1 (0.002) | 86.02 (0.06) | 0.013 (0.002) |

Table 5: Single-headed split-MNIST, FC-[256,256]. Accuracy and forgetting results.

| Method | memory size | 1 | | 5 | | 50 | |
|---|---|---|---|---|---|---|---|
| | memory | accuracy | forgetting | accuracy | forgetting | accuracy | forgetting |
| multi-task | ✗ | 95.2 | - | - | - | - | - |
| Fine-tune | ✗ | 52.52 (5.24) | 0.41 (0.06) | - | - | - | - |
| EWC | ✗ | 56.48 (6.46) | 0.31 (0.05) | - | - | - | - |
| A-GEM | ✓ | 34.04 (7.10) | 0.23 (0.11) | 33.57 (6.32) | 0.18 (0.03) | 33.35 (4.52) | 0.12 (0.04) |
| ER-Reservoir | ✓ | 34.63 (6.03) | 0.79 (0.07) | 63.60 (3.11) | 0.42 (0.05) | 86.17 (0.99) | 0.13 (0.016) |
| NCCL + Ring | ✓ | 34.64 (3.27) | 0.55 (0.03) | 61.02 (6.21) | 0.207 (0.07) | 81.35 (8.24) | -0.03 (0.1) |
| NCCL+Reservoir | ✓ | 37.02 (0.34) | 0.509 (0.009) | 65.4 (0.7) | 0.16 (0.006) | 88.9 (0.28) | -0.125 (0.004) |

Table 6: Single-headed split-MNIST, FC-[400,400] and mem. size=500(50 / cls.). Accuracy and forgetting results.

| Method | accuracy |
|---|---|
| multi-task | 96.18 |
| Fine-tune | 50.9 (5.53) |
| EWC | 55.40 (6.29) |
| A-GEM | 26.49 (5.62) |
| ER-Reservoir | 85.1 (1.02) |
| CN-DPM | 93.23 |
| Gdumb | 91.9 (0.5) |
| NCCL + Reservoir | 95.15 (0.91) |

Table 7: Single-headed split-CIFAR10, full size Resnet-18 and mem. size=500(50 / cls.). Accuracy and forgetting results.

| Method | accuracy |
|---|---|
| iid-offline | 93.17 |
| iid-online | 36.65 |
| Fine-tune | 12.68 |
| EWC | 53.49 (0.72) |
| A-GEM | 54.28 (3.48) |
| GSS | 33.56 |
| Reservoir Sampling | 37.09 |
| CN-DPM | 41.78 |
| NCCL + Ring | 54.63 (0.76) |
| NCCL + Reservoir | 55.43 (0.32) |

Table 8: Permuted-MNIST (23 tasks 10000 examples per task), FC-[256,256] and Multi-headed split-CIFAR100, full size Resnet-18. Accuracies with different clipping rate on NCCL + Ring.

| $\beta_{max}$ | Permuted-MNIST | Split-CIFAR100 |
|---|---|---|
| 0.001 | 72.52(0.59) | 49.43(0.65) |
| 0.01 | 72.93(1.38) | 56.95(1.02) |
| 0.05 | 72.18(0.77) | 56.35(1.42) |
| 0.1 | 72.29(1.34) | 58.20(0.155) |
| 0.2 | 74.38(0.89) | 57.60(0.36) |
| 0.5 | 72.95(0.50) | 59.06(1.02) |
| 1 | 72.92(1.07) | 57.43(1.33) |
| 5 | 72.31(1.79) | 57.75(0.24) |

Table 9: Permuted-MNIST (23 tasks 10000 examples per task), FC-[256,256] and Multi-headed split-CIFAR100, full size Resnet-18. Training time.

| Methods | Training time [s] | |
|---|---|---|
| | Permuted-MNIST | Split-CIFAR100 |
| fine-tune | 91 | 92 |
| EWC | 95 | 159 |
| A-GEM | 180 | 760 |
| ER-Ring | 109 | 129 |
| ER-Reservoir | 95 | 113 |
| ORTHOG-SUBSPACE | 90 | 581 |
| NCCL+Ring | 167 | 248 |
| NCCL+Reservoir | 168 | 242 |

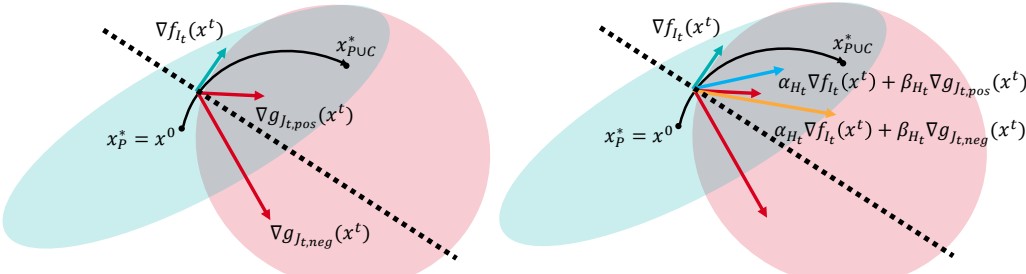

Figure 1: Geometric illustration of Non-Convex Continual Learning (NCCL). In the continual learning setting, the model parameter starts from the moderate local optimal point for the previously learned tasks $x_P^*$. Over $T$ iterations, we expect to reach a new optimal point $x_{P \cup C}^*$ which has a good performance on both previously learned and current tasks. In the $t$-th iteration, the model parameter $x^t$ encounters either $\nabla g_{J_t, pos}(x^t)$ or $\nabla g_{J_t, neg}(x^t)$. These two different cases indicates whether $\langle f_{I_t}(x^t), \nabla g_{J_t}(x^t) \rangle$ is positive or not. To prevent $x^t$ from escaping the feasible region, i.e., catastrophic forgetting, we impose a theoretical condition on learning rates for $f$ and $g$.

## B    THEORETICAL ANALYSIS

In this section, we provide the proofs of the results for nonconvex continual learning. We first start with the derivation of Equation 5 in Assumption 1.

### B.1    ASSUMPTION AND ADDITIONAL LEMMA

***Derivation of Equation 5.*** Recall that

$$|f_i(x) - f_i(y) - \langle \nabla f_i(y), x - y \rangle| \leq \frac{L}{2} \|x - y\|^2. \tag{17}$$

Note that $f_i$ is differentiable and nonconvex. We define a function $g(t) = f_i(y + t(x - y))$ for $t \in [0, 1]$ and an objective function $f_i$. By the fundamental theorem of calculus,

$$\int_0^1 g'(t) dt = f(x) - f(y). \tag{18}$$

By the property, we have

$$|f_i(x) - f_i(y) - \langle \nabla f_i(y), x - y \rangle|$$
$$= \left| \int_0^1 \langle \nabla f_i(y + t(x - y)), x - y \rangle dt - \langle \nabla f_i(y), x - y \rangle \right|$$
$$= \left| \int_0^1 \langle \nabla f_i(y + t(x - y)) - \nabla f_i(y), x - y \rangle dt \right|.$$

Using the Cauchy-Schwartz inequality,

$$\left| \int_0^1 \langle \nabla f_i(y + t(x - y)) - \nabla f_i(y), x - y \rangle dt \right|$$
$$\leq \left| \int_0^1 \|\nabla f_i(y + t(x - y)) - \nabla f_i(y)\| \cdot \|x - y\| dt \right|.$$

Since $f_i$ satisfies Equation 4, then we have

$$|f_i(x) - f_i(y) - \langle \nabla f_i(y), x - y \rangle|$$
$$\leq \left| \int_0^1 L\|y + t(x - y) - y\| \cdot \|x - y\| dt \right|$$
$$= L\|x - y\|^2 \left| \int_0^1 t dt \right|$$
$$= \frac{L}{2} \|x - y\|^2.$$

■

**Lemma B.1.** *Let $p = [p_1, \cdots p_D]$, $q = [q_1, \cdots, q_D]$ be two statistically independent random vectors with dimension $D$. Then the expectation of the inner product of two random vectors $\mathbb{E}[\langle p, q \rangle]$ is $\sum_{d=1}^{D} \mathbb{E}[p_d]\mathbb{E}[q_d]$.*

*Proof.* By the property of expectation,

$$\mathbb{E}[\langle p, q \rangle] = \mathbb{E}[\sum_{d=1}^{D} p_d q_d]$$
$$= \sum_{d=1}^{D} \mathbb{E}[p_d q_d]$$
$$= \sum_{d=1}^{D} \mathbb{E}[p_d]\mathbb{E}[q_d].$$

■

### B.2 PROOF OF MAIN RESULTS

We now show the main results of our work.

***Proof of Lemma 1***. To clarify the issue of $E_{M_t}[E_{I_t}[e_t|M_t]] = 0$, let us explain the details of constructing replay-memory as follows. We have considered episodic memory and reservoir sampling in the paper. We will first show the case of episodic memory by describing the sampling method for replay memory. We can also derive the case of reservoir sampling by simply applying the result of episodic memory.

**Episodic memory (ring buffer).** We divide the entire dataset of continual learning into the previous task $P$ and the current task $C$ on the time step $t = 0$. For the previous task $P$, the data stream of $P$ is i.i.d., and its sequence is random on every trial (episode). The trial (episode) implies that a continual learning agent learns from an online data stream with two consecutive data sequences of $P$ and $C$. Episodic memory takes the last data points of the given memory size $m$ by the First In First Out (FIFO) rule, and holds the entire data points until learning on $C$ is finished. Then, we note that $M_t = M_0$ for all $t \geq 0$ and $M_0$ is uniformly sampled from the i.i.d. sequence of $P$. By the law of total expectation, we derive $E_{M_0 \subset P}[E_{I_t}[\nabla f_{I_t}(x^t)|M_0]]$ for any $x^t$, $\forall t \geq 0$ as follows.

$$E_{M_0 \subset P}[E_{I_t}[\nabla f_{I_t}(x^t)|M_0]] = E_{M_0 \subset P}[\nabla f_{M_0}(x^t)].$$

We know that $M_0$ was uniformly sampled from $P$ on each trial before training on the current task $C$. Then, we take expectation with respect to every trial that implies the expected value over the memory distribution $M_0$. We get

$$E_{M_0 \subset P}[\nabla f_{M_0}(x^t)] = \nabla f(x^t)$$

for any $x^t$, $\forall t$. We can consider $\nabla f_{M_t}(x^t)$ as a sample mean of $P$ on every trial for any $x^t$, $\forall t \geq 0$. Although $x^t$ is constructed iteratively, the expected value of the sample mean for any $x^t$, $E_{M_0 \subset P}[\nabla f_{M_0}(x^t)]$ is also derived as $\nabla f(x^t)$.

**Reservoir sampling.** To clarify the notation for reservoir sampling first, we denote the expectation with respect to the history of replay memory $M_{[0:t]} = (M_0, \cdots, M_t)$ as $E_{M_{[0:t]}}$. This is the revised version of $E_{M_t}$. Reservoir sampling is the more tricky case than episodic memory, but $E_{M_{[0:t]}}[E_{I_t}[e_t|M_t]] = 0$ still holds. Suppose that $M_0$ is full of the data points from $P$ as the episodic memory is sampled and the mini-batch size from $C$ is 1 for simplicity. The reservoir sampling algorithm drops a data point in $M_{t-1}$ and replaces the dropped data point with a data point in the current mini-batch from $C$ with probability $p = m/n$, where $m$ is the memory size and $n$ is the number of visited data points so far. The exact pseudo-code for reservoir sampling is described in [1]. The replacement procedure uniformly chooses the data point which will be dropped. We can

also consider the replacement procedure as follows. The memory $M_t$ for $P$ is reduced in size 1 from $M_{t-1}$, and the replaced data point $d_C$ from $C$ contributes in terms of $\nabla g_{d_C}(x^t)$ if $d_C$ is sampled from the replay memory. Let $M_{t-1} = [d_1, \cdots, d_{|M_{t-1}|}]$ where $|\cdot|$ denotes the cardinality of the memory. The sample mean of $M_{t-1}$ is given as

$$\nabla f_{M_{t-1}}(x^{t-1}) = \frac{1}{|M_{t-1}|} \sum_{d_i} \nabla f_{d_i}(x^{t-1}). \tag{19}$$

By the rule of reservoir sampling, we assume that the replacement procedure reduces the memory from $M_{t-1}$ to $M_t$ with size $|M_{t-1}| - 1$ and the set of remained upcoming data points $C_t \in C$ from the current data stream for online continual learning is reformulated into $C_{t-1} \cup [d_C]$. Then, $d_C$ can be resampled from $C_{t-1} \cup [d_C]$ to be composed of the minibatch of reservoir sampling with the dfferent probability. However, we ignore the probability issue now to focus on the effect of replay-memory on $\nabla f$. Now, we sample $M_t$ from $M_{t-1}$, then we get the random vector $\nabla f_{M_t}(x^t)$ as

$$\nabla f_{M_t}(x^t) = \frac{1}{|M_t|} \sum_{j=1}^{|M_{t-1}|} W_{ij} \nabla f_{d_j}(x^t), \tag{20}$$

where the index $i$ is uniformly sampled from $i \sim [1, \cdots, |M_{t-1}|]$, and $W_{ij}$ is the indicator function that $W_{ij}$ is 0 if $i = j$ else 1.

The above description implies the dropping rule, and $M_t$ can be considered as an uniformly sampled set with size $|M_t|$ from $M_{t-1}$. There could also be $M_t = M_{t-1}$ with probability $1 - p = 1 - m/n$. Then the expectation of $\nabla f_{M_t}(x^t)$ given $M_{t-1}$ is derived as

$$E_{M_t}[\nabla f_{M_t}(x^t)|M_{t-1}] = p \left( \frac{1}{|M_{t-1}|} \sum_i^{|M_{t-1}|} \frac{1}{|M_t|} \sum_{j=1}^{|M_{t-1}|} W_{ij} \nabla f_{d_j}(x^t) \right) + (1-p) \left( \nabla f_{M_{t-1}}(x^t) \right)$$
$$= \nabla f_{M_{t-1}}(x^t).$$

When we consider the mini-batch sampling, we can formally reformulate the above equation as

$$E_{M_t \sim p(M_t|M_{t-1})} \left[ E_{I_t \subset M_t} \left[ \nabla f_{I_t}(x^t)|M_t \right] |M_{t-1} \right] = \nabla f_{M_{t-1}}(x^t). \tag{21}$$

Now, we apply the above equation recursively. Then,

$$E_{M_1 \sim p(M_1|M_0)} \left[ \cdots E_{M_t \sim p(M_t|M_{t-1})} \left[ E_{I_t \subset M_t} \left[ \nabla f_{I_t}(x^t)|M_t \right] |M_{t-1} \right] \cdots |M_0 \right] = \nabla f_{M_0}(x^t). \tag{22}$$

Similar to episodic memory, $M_0$ is uniformly sampled from $P$. Therefore, we conclude that

$$E_{M_0, \cdots, M_t}[\nabla f_{M_t}(x^t)] = \nabla f(x^t) \tag{23}$$

by taking expectation over the history $M_{[0:t]} = (M_1, M_2, \cdots, M_t)$.

Note that taking expectation iteratively with respect to the history $M_{[t]}$ is needed to compute the expected value of gradients for $M_t$. However, the result $E_{M_0, \cdots, M_t}[E_{I_t}[e_t|M_t]] = 0$ still holds in terms of expectation. ∎

***Proof of Lemma 2***. We analyze the convergence of nonconvex continual learning with replay memory here. Recall that the gradient update is the following

$$x^{t+1} = x^t - \alpha_{H_t} \nabla f_{I_t}(x^t) - \beta_{H_t} \nabla g_{J_t}(x^t)$$

for all $t \in \{1, 2, \cdots, T\}$. Let $e_t = \nabla f_{I_t}(x^t) - \nabla f(x^t)$. Since we assume that $f$, $g$ is $L$-smooth, we have the following inequality by applying Equation 5:

$$f(x^{t+1}) \leq f(x^t) + \langle \nabla f(x^t), x^{t+1} - x^t \rangle + \frac{L}{2}\|x^{t+1} - x^t\|^2$$

$$= f(x^t) - \langle \nabla f(x^t), \alpha_{H_t}\nabla f_{I_t}(x^t) + \beta_{H_t}\nabla g_{J_t}(x^t) \rangle + \frac{L}{2}\|\alpha_{H_t}\nabla f_{I_t}(x^t) + \beta_{H_t}\nabla g_{J_t}(x^t)\|^2$$

$$= f(x^t) - \alpha_{H_t}\langle \nabla f(x^t), \nabla f_{I_t}(x^t) \rangle - \beta_{H_t}\langle \nabla f(x^t), \nabla g_{J_t}(x^t) \rangle$$
$$+ \frac{L}{2}\alpha_{H_t}^2\|\nabla f_{I_t}(x^t)\|^2 + \frac{L}{2}\beta_{H_t}^2\|\nabla g_{J_t}(x^t)\|^2 + L\alpha_{H_t}\beta_{H_t}\langle \nabla f_{I_t}(x^t), \nabla g_{J_t}(x^t) \rangle$$

$$= f(x^t) - \alpha_{H_t}\langle \nabla f(x^t), \nabla f(x^t) \rangle - \alpha_{H_t}\langle \nabla f(x^t), e_t \rangle - \beta_{H_t}\langle \nabla f_{I_t}(x^t), \nabla g_{J_t}(x^t) \rangle + \beta_{H_t}\langle \nabla g_{J_t}(x^t), e_t \rangle$$
$$+ \frac{L\alpha_{H_t}^2}{2}\|\nabla f(x^t)\|^2 + L\alpha_{H_t}^2\langle \nabla f(x^t), e_t \rangle + \frac{L\alpha_{H_t}^2}{2}\|e_t\|^2 + \frac{L\beta_{H_t}^2}{2}\|\nabla g_{J_t}(x^t)\|^2 + L\alpha_{H_t}\beta_{H_t}\langle \nabla f_{I_t}(x^t), \nabla g_{J_t}(x^t) \rangle$$

$$= f(x^t) - \left(\alpha_{H_t} - \frac{L}{2}\alpha_{H_t}^2\right)\|\nabla f(x^t)\|^2 + \frac{L}{2}\beta_{H_t}^2\|\nabla g_{J_t}(x^t)\|^2 - \beta_{H_t}(1 - \alpha_{H_t}L)\langle \nabla f_{I_t}(x^t), \nabla g_{J_t}(x^t) \rangle$$

$$+ \left(L\alpha_{H_t}^2 - \alpha_{H_t}\right)\langle \nabla f(x^t), e_t \rangle + \beta_{H_t}\langle \nabla g_{J_t}(x^t), e_t \rangle + \frac{L}{2}\alpha_{H_t}^2\|e_t\|^2. \tag{24}$$

To show the proposed theoretical convergence analysis of nonconvex continual learning, we define the catastrophic forgetting term $\Gamma_t$ and the overfitting term $B_t$ as follows:

$$B_t = (L\alpha_{H_t}^2 - \alpha_{H_t})\langle \nabla f(x^t), e_t \rangle + \beta_{H_t}\langle \nabla g_{J_t}(x^t), e_t \rangle,$$

$$\Gamma_t = \frac{\beta_{H_t}^2 L}{2}\|\nabla g_{J_t}(x^t)\|^2 - \beta_{H_t}(1 - \alpha_{H_t}L)\langle \nabla f_{I_t}(x^t), \nabla g_{J_t}(x^t) \rangle.$$

Then, we can rewrite Equation 24 as

$$f(x^{t+1}) \leq f(x^t) - \left(\alpha_{H_t} - \frac{L}{2}\alpha_{H_t}^2\right)\|\nabla f(x^t)\|^2 + \Gamma_t + B_t + \frac{L}{2}\alpha_{H_t}^2\|e_t\|^2. \tag{25}$$

We first note that $B_t$ is dependent of the error term $e_t$ with the batch $I_t$. In the continual learning step, an training agent cannot access $\nabla f(x^t)$, then we cannot get the exact value of $e_t$. Furthermore, $\Gamma_t$ is dependent of the gradients $\nabla f_{I_t}(x^t), \nabla g_{I_t}(x^t)$ and the learning rates $\alpha_{H_t}, \beta_{H_t}$.

Taking expectations with respect to $I_t$ on both sides given $J_t$, we have

$$\mathbb{E}_{I_t}\left[f(x^{t+1})\right] \leq \mathbb{E}_{I_t}\left[f(x^t) - \left(\alpha_{H_t} - \frac{L}{2}\alpha_{H_t}^2\right)\|\nabla f(x^t)\|^2 + \Gamma_t + B_t + \frac{L}{2}\alpha_{H_t}^2\|e_t\|^2\Big|J_t\right]$$

$$\leq \mathbb{E}_{I_t}\left[f(x^t) - \left(\alpha_{H_t} - \frac{L}{2}\alpha_{H_t}^2\right)\|\nabla f(x^t)\|^2 + \frac{L}{2}\alpha_{H_t}^2\|e_t\|^2\right] + \mathbb{E}_{I_t}\left[\Gamma_t + B_t\Big|J_t\right].$$

Now, taking expectations over the whole stochasticity we obtain

$$\mathbb{E}\left[f(x^{t+1})\right] \leq \mathbb{E}\left[f(x^t) - \left(\alpha_{H_t} - \frac{L}{2}\alpha_{H_t}^2\right)\|\nabla f(x^t)\|^2 + \Gamma_t + B_t + \frac{L}{2}\alpha_{H_t}^2\|e_t\|^2\right].$$

Rearranging the terms and assume that $\frac{1}{1 - L\alpha_{H_t}/2} > 0$, we have

$$\left(\alpha_{H_t} - \frac{L}{2}\alpha_{H_t}^2\right)\mathbb{E}\|\nabla f(x^t)\|^2 \leq \mathbb{E}\left[f(x^t) - f(x^{t+1}) + \Gamma_t + B_t + \frac{L}{2}\alpha_{H_t}^2\|e_t\|^2\right]$$

and

$$\mathbb{E}\|\nabla f(x^t)\|^2 \leq \mathbb{E}\left[\frac{1}{\alpha_{H_t}(1 - \frac{L}{2}\alpha_{H_t})}\left(f(x^t) - f(x^{t+1}) + \Gamma_t + B_t\right) + \frac{\alpha_{H_t}L}{2(1 - \frac{L}{2}\alpha_{H_t})}\|e_t\|^2\right]$$

$$\leq \mathbb{E}\left[\frac{1}{\alpha_{H_t}(1 - \frac{L}{2}\alpha_{H_t})}\left(f(x^t) - f(x^{t+1}) + \Gamma_t + B_t\right) + \frac{\alpha_{H_t}L}{2(1 - \frac{L}{2}\alpha_{H_t})}\sigma_f^2\right].$$

$\blacksquare$

***Proof of Theorem 1***. Suppose that the learning rate $\alpha_{H_t}$ is a constant $\alpha = c/\sqrt{T}$, for $c > 0$, $1 - \frac{L}{2}\alpha = \frac{1}{A} > 0$. Then, by summing Equation 10 from $t = 0$ to $T - 1$, we have

$$
\begin{aligned}
\min_t \mathbb{E}\|\nabla f(x^t)\|^2 &\leq \frac{1}{T} \sum_{t=0}^{T-1} \mathbb{E}\|\nabla f(x^t)\|^2 \\
&\leq \frac{1}{1 - \frac{L}{2}\alpha} \left( \frac{1}{\alpha T} \left( f(x^0) - f(x^T) + \sum_{t=0}^{T-1} (\mathbb{E}\left[B_t + \Gamma_t\right]) \right) + \frac{L}{2}\alpha\sigma_f^2 \right) \\
&= \frac{1}{1 - \frac{L}{2}\alpha} \left( \frac{1}{c\sqrt{T}} \left( \Delta_f + \sum_{t=0}^{T-1} (\mathbb{E}\left[B_t + \Gamma_t\right]) \right) + \frac{Lc}{2\sqrt{T}}\sigma_f^2 \right) \\
&= \frac{A}{\sqrt{T}} \left( \frac{1}{c} \left( \Delta_f + \sum_{t=0}^{T-1} \mathbb{E}\left[B_t + \Gamma_t\right] \right) + \frac{Lc}{2}\sigma_f^2 \right).
\end{aligned}
\tag{26}
$$

We note that a batch $I_t$ is sampled from a memory $M_t \subset M$ which is a random vector whose element is a datapoint $d \in P \cup C$. Then, taking expectation over $I_t \subset M_t \subset P \cup C$ implies that $\mathbb{E}[B_t] = 0$. Therefore, we get the minimum of expected square of the norm of gradients as follows:

$$
\min_t \mathbb{E}\|\nabla f(x^t)\|^2 \leq \frac{A}{\sqrt{T}} \left( \frac{1}{c} \left( \Delta_f + \sum_{t=0}^{T-1} \mathbb{E}[\Gamma_t] \right) + \frac{Lc}{2}\sigma_f^2 \right).
$$

∎

***Proof of Lemma 3***. By the assumption, it is equivalent to update on $M \cup C$. Then, the non-convex finite sum optimization is given as

$$
\min_{x \in \mathbb{R}^d} g(x) = \frac{1}{n_g + |M|} \sum_{i \in M \cup C} g_i(x),
\tag{27}
$$

where $g_i$ is the same function as $f_i$, and $|M|$ is the number of elements in $M$. This problem can be solved by a simple SGD algorithm (Reddi et al., 2016b). Thus, we have

$$
\min_t \mathbb{E}\|\nabla g(x^t)\|^2 \leq \sqrt{\frac{2\Delta_g L}{T}}\sigma_g.
\tag{28}
$$

∎

**Lemma B.2.** *Let an upper bound $\beta > \beta_{H_t} > 0$. For the worst case, the expectation of summing the catastrophic forgetting term over iterations $T$ is*

$$
\sum_{t=0}^{T-1} \Gamma_t = O(T).
$$

*Proof.* First, we derive the rough upper bound of $E[\Gamma_t]$:

$$
E[\Gamma_t] = E\left[ \frac{\beta_{H_t}^2 L}{2}\|\nabla g_{J_t}(x^t)\|^2 - \beta_{H_t}(1 - \alpha_{H_t}L)\langle \nabla f_{I_t}(x^t), \nabla g_{J_t}(x^t) \rangle \right]
\tag{29}
$$

$$
\leq E\left[ \frac{\beta_{H_t}^2 L}{2}\|\nabla g_{J_t}(x^t)\|^2 + \beta_{H_t}(1 - \alpha_{H_t}L)\|\nabla f_{I_t}(x^t)\|\|\nabla g_{J_t}(x^t)\| \right]
\tag{30}
$$

$$
= O\left( E\left[ \frac{\beta^2 L}{2}\|\nabla g_{J_t}(x^t)\|^2 \right] \right)
\tag{31}
$$

where $\|\nabla g_{J_t}(x^t)\| \geq \|\nabla f_{I_t}(x^t)\|$.

In addition, the supremum of the variance of the mini-batch gradient $\nabla g_{J_t}(x^t)$ is derived as

$$\sup_x E\|\nabla g_{J_t}(x^t) - \nabla g(x^t)\|^2 = \sup_x \frac{n_g - b_g}{(n_g - 1)b_g} \cdot \frac{1}{n_g} \sum_{j=1}^{n_g} \|\nabla g_j(x^t) - \nabla g(x^t)\|^2 \tag{32}$$

$$= \frac{n_g - b_g}{(n_g - 1)b_g} \sigma_g^2, \tag{33}$$

where $n_g$ and $b_g$ denotes the size of $C$ and minibatch $J_t$, respectively. The detailed derivation is shown in [1]. By the triangular inequality, we get

$$E\|g_{J_t}(x^t)\|^2 \leq E\|g_{J_t}(x^t) - g(x^t)\|^2 + E\|g(x^t)\|^2 \tag{34}$$

$$\leq E\|g(x^t)\|^2 + \frac{n_g - b_g}{(n_g - 1)b_g} \sigma_g^2. \tag{35}$$

By plugging the above equation into $E[\Gamma_t]$, we conclude that

$$E[\Gamma_t] = O\left( E\left[ \frac{\beta^2 L}{2} \|\nabla g_{J_t}(x^t)\|^2 \right] \right) \tag{36}$$

$$= O\left( E\left[ \frac{\beta^2 L}{2} \|\nabla g(x^t)\|^2 \right] + \frac{\beta^2 L(n_g - b_g)}{2(n_g - 1)b_g} \sigma_g^2 \right). \tag{37}$$

The sum of catastrophic forgetting term $\sum \Gamma_t$ is corrected as $\sum E[\Gamma_t]$. We use the technique for summing up in the proof of Theorem 1, then the cumulative sum of catastrophic forgetting term is derived as

$$\sum_{t=0}^{T-1} E[\Gamma_t] = \sum_{t=0}^{T-1} \frac{\beta^2 L}{2} O\left( E\left[ \|\nabla g(x^t)\|^2 \right] + \frac{(n_g - b_g)}{(n_g - 1)b_g} \sigma_g^2 \right) \tag{38}$$

$$\leq \frac{\beta^2 L}{2} \sum_{t=0}^{T-1} O\left( \frac{1}{\beta} \left[ g(x^t) - g(x^{t+1}) \right] + \frac{L\beta}{2} \sigma_g^2 + \frac{(n_g - b_g)}{(n_g - 1)b_g} \sigma_g^2 \right) \tag{39}$$

$$\leq \frac{\beta^2 L}{2} O\left( \frac{1}{\beta} \Delta_g + \frac{TL\beta}{2} \sigma_g^2 + \frac{T(n_g - b_g)}{(n_g - 1)b_g} \sigma_g^2 \right) \tag{40}$$

$$= O\left( \beta\Delta_g + \sigma_g^2 \left( \frac{L\beta^3}{2} + \frac{(n_g - b_g)\beta^2}{(n_g - 1)b_g} \right) T \right). \tag{41}$$

Rearranging the above equation, we get

$$\sum_{t=0}^{T-1} E[\Gamma_t] = O\left( \sigma_g^2 \left( \frac{L\beta^3}{2} + \frac{(n_g - b_g)\beta^2}{(n_g - 1)b_g} \right) T + \beta\Delta_g \right). \tag{42}$$

Therefore, we can write $\sum_{t=0}^{T-1} E[\Gamma_t] = O(T)$. We note that the rough upper bound of $\sum \Gamma_t$ increases monotonically with training step as in the previous result in the paper.

Now we provide the rigorous derivation of the convergence rate by the result of Corollary 1 as follows:

$$O\left( \sigma_g^2 \left( \frac{L\beta^3}{2} + \frac{(n_g - b_g)\beta^2}{(n_g - 1)b_g} \right) \sqrt{T} + \frac{\beta\Delta_g}{\sqrt{T}} \right) = O(\sqrt{T}). \tag{43}$$

This result is obtained by dividing $\sum E[\Gamma_t]$ by $\sqrt{T}$ as in the proof of Thm. 1.

■

On the other hand, $\Gamma_t$ can be negative when $\langle \nabla f_{I_t}(x^t), \nabla g_{J_t}(x^t) \rangle > 0$. It implies that the cumulative sum of $\Gamma_t$ does not increase monotonically. Therefore, for some large number $N$, we can denote the cumulative sum of $\Gamma_t$ over the finite steps $T$ as follows:

$$\sum_{t=0}^{T-1} \Gamma_t \leq N = O(1). \tag{44}$$

**Proof of Corollary 1**. To formulate the IFO calls, Recall that $T(\epsilon)$

$$T(\epsilon) = \min\{T: \min \mathbb{E}\|\nabla f(x^t)\|^2 \le \epsilon\}.$$

Additionally, a single IFO call is invested in calculating each step. As seen in Theorem 1, NCCL has a convergence rate of

$$O\left(\frac{\sum_{t=0}^{T-1}\Gamma_t}{\sqrt{T}}\right). \tag{45}$$

We note that the convergence rate for the worst case is

$$O\left(\sqrt{T}\right), \tag{46}$$

where the given model diverges on the convergence of $f(x)$. Then, IFO calls are denoted as $\infty$.

For the case of Equation 44, we obtain the convergence rate $O(1/\sqrt{T})$. Thus we get $O(1/\epsilon^2)$ in this case.

∎

## C   DERIVATION OF EQUATIONS IN SECTION 3.4

**Derivation for A-GEM**    Let the surrogate $\nabla \tilde{g}_{J_t}(x^t)$ as

$$\nabla \tilde{g}_{J_t}(x^t) = \nabla g_{J_t}(x^t) - \left\langle \frac{\nabla f_{I_t}(x^t)}{\|\nabla f_{I_t}(x^t)\|}, \nabla g_{J_t}(x^t) \right\rangle \frac{\nabla f_{I_t}(x^t)}{\|\nabla f_{I_t}(x^t)\|}, \tag{47}$$

where $\alpha_{H_t} = \alpha(1 - \frac{\langle \nabla f_{I_t}(x^t), \nabla g_{J_t}(x^t)\rangle}{\|\nabla f_{I_t}(x^t)\|^2})$ and $\beta_{H_t} = \alpha$ for Equation 7.

Then, we have

$$
\begin{aligned}
\mathbb{E}[\Gamma_t] &= \mathbb{E}\left[\frac{\beta_{H_t}^2 L}{2}\|\nabla \tilde{g}_{J_t}(x^t)\|^2 - \beta_{H_t}\langle \nabla f_{I_t}(x^t), \nabla \tilde{g}_{J_t}(x^t)\rangle\right] \\
&= \mathbb{E}\left[\frac{\beta_{H_t}^2 L}{2}\left(\|\nabla g_{J_t}(x^t)\|^2 - 2\frac{\langle \nabla f_{I_t}(x^t), \nabla g_{J_t}(x^t)\rangle^2}{\|\nabla f_{I_t}(x^t)\|^2} + \frac{\langle \nabla f_{I_t}(x^t), \nabla g_{J_t}(x^t)\rangle^2}{\|\nabla f_{I_t}(x^t)\|^2}\right) - \beta_{H_t}\langle \nabla f_{I_t}(x^t), \nabla \tilde{g}_{J_t}(x^t)\rangle\right] \\
&= \mathbb{E}\left[\frac{\beta_{H_t}^2 L}{2}\left(\|\nabla g_{J_t}(x^t)\|^2 - \frac{\langle \nabla f_{I_t}(x^t), \nabla g_{J_t}(x^t)\rangle^2}{\|\nabla f_{I_t}(x^t)\|^2}\right) - \beta_{H_t}\left(\langle \nabla f_{I_t}(x^t), \nabla g_{J_t}(x^t)\rangle - \langle \nabla f_{I_t}(x^t), \nabla g_{J_t}(x^t)\rangle\right)\right] \\
&= \mathbb{E}\left[\frac{\beta_{H_t}^2 L}{2}\left(\|\nabla g_{J_t}(x^t)\|^2 - \frac{\langle \nabla f_{I_t}(x^t), \nabla g_{J_t}(x^t)\rangle^2}{\|\nabla f_{I_t}(x^t)\|^2}\right)\right]. 
\end{aligned} \tag{48}
$$

Note that this result is smaller than the original $\mathbb{E}[\Gamma_t]$.

**Derivation of optimal $\Gamma_t^*$ and $\beta_{H_t}^*$**    For a fixed learning rate $\alpha$, we have

$$
\begin{aligned}
0 = \frac{\partial \mathbb{E}[\Gamma_t]}{\partial \beta_{H_t}} &= \mathbb{E}\left[\frac{\partial \Gamma_t}{\partial \beta_{H_t}}\right] \\
&= \mathbb{E}\left[\beta_{H_t} L\|\nabla g_{J_t}(x^t)\| - (1 - \alpha L)\langle \nabla f_{I_t}(x^t), \nabla g_{J_t}(x^t)\rangle\right].
\end{aligned}
$$

Thus, we obtain

$$\beta_{H_t}^* = \frac{(1 - \alpha_{H_t} L)\langle \nabla f_{I_t}(x^t), \nabla g_{J_t}(x^t)\rangle}{L\|\nabla g_{J_t}(x^t)\|^2},$$

$$\Gamma_t^* = -\frac{(1 - \alpha_{H_t} L)\langle \nabla f_{I_t}(x^t), \nabla g_{J_t}(x^t)\rangle}{2L\|\nabla g_{J_t}(x^t)\|^2}.$$

## D   OVERFITTING TO REPLAY MEMORY

In the main text, we discussed a theoretical convergence analysis of continual learning for a smooth nonconvex finite-sum optimization problems. The practical continual learning tasks have the restriction on full access to the entire data points of previously learned tasks. Unlike taking expectation

over $I_t \sim M$ and $M \sim P \cup C$, we have to compute on the given memory in the practical scenario. Then, we note that $\mathbb{E}[B_t|M] \neq 0$.

Now we rewirte Equation 26 for the worst case as follows.

$$T \sup \|\nabla f(x)\|^2 \leq \frac{1}{\alpha(1 - \alpha L/2)} \left( \Delta_f + \sum (B_t + C_t) + \frac{L}{2} \alpha^2 \sigma_f^2 \right) \tag{49}$$

$$\sup \|\nabla f(x)\|^2 \leq \frac{A}{\sqrt{T}} \left( \frac{1}{c} \left( \Delta_f + \sum (B_t + C_t) \right) + \frac{Lc}{2} \sigma_f^2 \right). \tag{50}$$

We note that $\sum B_t$ is a random variable, which is unpredictible, and choosing $\nabla f_M(t) = \nabla f(x^t)$ over entire period is impossible. Then, the cumulative sum of $B_t$ is increasing over $T$. Therefore, we conclude that for the overfitting to memory degrades the convergence rate of NCCL empirically.

