# OpenReview forum: "On the Convergence of Nonconvex Continual Learning with Adaptive Learning Rate"
_ICLR.cc/2022/Conference — ICLR 2022 Submitted_

### Official Review · Reviewer_qew6 · 2021-10-17

**Correctness:** 4
**Technical Novelty And Significance:** 3
**Empirical Novelty And Significance:** 3
**Recommendation:** 8
**Confidence:** 3

**Main Review:**

Pros:
 - The paper is well-written, clear, and straightforward.
 - The results are new and interesting
 - Theoretical analysis is well-conducted

Cons:
 - The guarantees not surprising and maybe a bit incremental
 - Proof techniques are standard

Other than the small issue in Lemma 2: note that \alpha is allowed to be 0 but then (10) is not well-defined, so it should be strictly larger than 0, I did not find any problems in the proofs in the appendix or in the mathematical writing.

**Summary Of The Paper:**

The paper proposes mini-batch stochastic gradient method of the framework of nonconvex memory-based continual learning, and conducts  a theoretical convergence analysis. Additionally, the paper studies catastrophic forgetting and outfitting in the context of the proposed approach.



**Summary Of The Review:**

 The paper is interesting, provides new insights and theoretical guarantees, and in my opinion should be accepted for publication.

---

> ### Author Response · Authors · 2021-11-23
> **Response to qew6**
>
> Thank you for the thoughtful review and the insight on our paer.
>
>
> We have revised the condition of learning rate in Equation 10 for continual learning.
> It is true that $\alpha$ should  be strictly larger than 0 to make RHS be finite.
> When $\alpha=0$, it is a memoryless setting for learning the current task $C$.
> With the naive nonconvex SGD, we cannot gurantee the convergence on the previous task $f$.
> Therefore, $\alpha$ is to be strictly larger than 0 to analyze the convergence rate of nonconvex continual learning.

---

### Official Review · Reviewer_TD2U · 2021-10-29

**Correctness:** 3
**Technical Novelty And Significance:** 3
**Empirical Novelty And Significance:** 3
**Recommendation:** 5
**Confidence:** 3

**Main Review:**

My detailed comments are provided as below.

Strength:

1, Continual learning is a timely and important topic for modern machine learning. This paper provides a good step to understand the optimization properties of this direction. The analysis captures something that has not been encountered in conventional optimization problem, such as the overfitting error term and the catastrophic forgetting term, although they do not provide a quantitative characterization on the catastrophic forgetting error.

2. The design of the adaptive learning rates motivated by the analysis is interesting. It can help to reduce the catastrophic forgetting error and seem to perform well in the experiments.

Weakness:

1. This paper is not well written. For example, it is really hard for me to understand Lemma 1, which is an important result to show how the overfitting error vanish when the initial memory setup $M_0$ is properly selected, because the authors do not do a good job in explaining the episodic memory and the replay memory with sample dropping. Even after I carefully read their proofs, I figure out in the episodic memory case, $M_t=M_0$ for all $t$, whereas for the dropping case, $M_{t}$ samples from previous $M_{t-1}$ with one random sample dropping. Given such settings, it is not very surprising to me to get such a result.

2. The authors do not provide a quantitative characterization on the overfitting error (for the general case without the initialization as in Lemma 1)  and the catastrophic forgetting error. For example, it is more interesting to characterize, even under some extra assumptions, how and when such errors can be bounded. So far I do not find such results.

3. I am wondering whether the authors can provide some figures to show whether their proposed algorithms indeed achieve a faster convergence rate in terms of the iteration number and the catastrophic forgetting error.

4. Since in practice, there is a long sequence of tasks (beyond two as studied in this paper) for continual learning, I am wondering whether the analysis here can be further generalized to this more general case. This is because based on my knowledge, catastrophic forgetting error may also depend on the how far between two tasks.

5. Some typos:
1) page 4 line 7: "the the replay memory with..." ->  "the replay memory with..."
2) page 8 last row but three: "Table ??": no quotation number.















**Summary Of The Paper:**

This paper provides a convergence rate characterization for a continual learning problem where the objective functions (for two consecutive tasks) are generally nonconvex and have a finite-sum form over data samples (which belong to current and previous tasks). The analysis uses the tools from nonconvex optimization, with a goal to find a stationary point for current and previous tasks. Under the setting of continual learning, the authors consider the standard replay memory based method, focusing on both the episodic memory and the replay memory with sample dropping. For the analysis, the authors first address two important types of errors: gradient estimation bias at time t and the catastrophic forgetting error. They show that for these two memory based methods, with a good initialization of replay memory, the gradient estimation error vanishes. They further show that in term of the convergence rate of previous task, i.e., $f$ in its context, the catastrophic forgetting error seems to be inevitable and can be even unbounded. To address this error, they further propose some adaptive learning rate schemes, which show some effectiveness in several experiments.

**Summary Of The Review:**

Overall, I feel this paper provides some initial steps to understand the convergence of continual learning, but given some weakness I list above, I am slightly negative about it. However, I am open to increase my score based on the authors' feedback and other reviewers' comments.

---

> ### Author Response · Authors · 2021-11-23
> **Response to Reviewer TD2U**
>
> Thank you for your thoughtful comments and suggestions for how to improvethe paper.
> We have added more discussion on the revised version to answer your concerns.
>
> > hard to understand Lemma 1, show how the overfitting vanish when the initial memory setup $M_0$ is properly selected.
>
>
> In view of expectation over the randomness of $M_0$, we note that considering a random choice of $M_0$ allows us to analyze the convergence on $f(x)$, even if $M_0$ cannot access the all datapoints in $P$ to compute $\lVert \nabla f(x^t) \rVert^2.$
>
> In the below of Lemma 2, we also note that the individual trial with a randomly given $M_0$ still has the degradation of upper bound by the overfitting term $B_t$, although its total expectation over the whole possible trials $\mathbb{E}[B_t]$ is zero.
>
> Therefore, the overfitting bias on memory does not vanish on each trial, and we cannot reduce the degradation by $B_t$.
> This is because we cannot access the whole datapoints in $P$ to compute $e_t$ and $\nabla f(x^t)$.
> We try to alleviate the overfitting bias by applying the small size of batch on both $I_t$ and $J_t$.
> It is known that small batch can reduce the overfitting bias by reducing the sharpness [1].
>
> [1] Keskar, Nitish Shirish, et al. "On large-batch training for deep learning: Generalization gap and sharp minima." ICLR 2017.
>
> > do not provide a quantitative characterization on catastrophic forgetting error and explanation of the quantitavie characterization on the overfitting error.
>
> To provide a quantitative measures on catastrophic forgetting and overfitting, we are running the additional experiments.
> However, it is hard to compute the gradient of entire datasets on a single phase $\nabla f(x^t)$ for the overfiting bias.
> It takes more time than we expect.
> We will upload this result as soon as possible.
>
>
>
> > a long seqeunce of tasks, can be generalized to this more general case?
>
> Yes. We can generalize our result for the long sequence of tasks.
> In Alogithm 1, we have already described the procedure with multiple new tasks by adding $\cup J_t$ into $P$.
> For the long seqeunce of tasks, we can apply our theoretical result iteratively with updating $P$ and $M$.
> Then, the cumulative sum of $\Gamma_t$ has much more elements by applying on the general case.

---

> > ### Author Response · Authors · 2021-11-30
> > **Additional response**
> >
> > We provide more detailed responses on your comments and suggestions.
> >
> >
> > > Additional comments on Lemma 1.
> >
> > We agree that Lemma 1 is a simple proposition that allows to vanish the effect of overfitting term $B_t$ when analyzing the expected value of square of norm of gradients.
> > However, the uniformly sampled $M_0$ by Lemma 1 allows us to compute the expectation over P by computing $E_{P} [ E_{M_0 \sim P} [ \cdot ]]$.
> > Unlike the general SGD, at the gradient update step, the algorithm cannot sample a data point from $P$, but it can access data points in the memory $M$ which is the subset of $P$.
> > This fact makes the continual learning scenario hard to be theoretically analyzed on the convergence of $P$, which is the previous tasks.
> > Therefore, Lemma 1 provides the quantitatively improved performance on catastrophic forgetting regardless of the choice of memory, and also allows us to compute the expectation over $P$ by the random sampling of $M_0$ to analyze the convergence on $P$, which is not trivial.

---

> > > ### Author Response · Authors · 2021-12-01
> > > **We provide the quantitative analysis of catastrophic forgetting and overfitting to memory above.**
> > >
> > > Thank you for your thoughtful feedback!
> > > As you suggested in your review, and we mentioned in the response, we have successfully conducted the additional experiments on quantitative analysis of the cumulative sum of catstrophic forgetting term $\sum_{t} \Gamma_t$ among baslines.
> > >
> > > We note that minimizing the cumulative sum of catastrophic forgetting term as we proposed in Equation (14) of our paper, is the key contribution to optimizing the forgetting metric.
> > > The cumulative sum is also highly correlated with the test average accuracy, so the our NCCL variants show the reasonable performances when we compare to other state-of-the art baselines.
> > > The reason that NCCL + Ring with the minimum cumulative sum shows the best score in terms of test average accuracy is that the proposed algorithm is to optimize the convergence on $f(x)$, and the algorithm slightly suffers from learning the current tasks than ER-Ring.
> > > We think that we can overcome this problem by increasing the memory size to get more stable estimation on the expectation of $\Gamma_t$ (larger memory), or the future research to analyze the faster convergence on $g(x)$.
> > >
> > > ################################################################
> > >
> > > We additionally update the result and discussion on overfitting to memory above.
> > >
> > > Please let us know if you have any further questions.
> > >
> > > #################################################################
> > >
> > > Finally, we will upload the visulization plot of the above result to make our result easy to follow in the next version.
> > > Please let us know if you have an additional concerns about our feedback or our paper.

---

### Official Review · Reviewer_UMAT · 2021-11-01

**Correctness:** 3
**Technical Novelty And Significance:** 2
**Empirical Novelty And Significance:** 3
**Recommendation:** 3
**Confidence:** 4

**Main Review:**

Strengths:

1. The authors propose a theoretical analysis of the memory based lifelong learning algorithms which is an important problem since currently most of the memory based lifelong learning algorithms lack  theoretical  guarantee.

2. The authors make a good connection to existing memory based lifelong learning algorithms such as GEM and A-GEM and discussed their drawbacks.

Weaknesses:

1. The writing of the paper needs improvement. There are many unclear sentences and typos in the paper. The notations are also confusing.

2. The main issues of paper are the experimental results. In table 1, NCCL + Reservoir and NCCL + Ring underperforms ER-Ring, could the authors give some discussion on this?

3. The authors argue that "the adaptive learning rate scheme is to prevent catastrophic forgetting", but the average accuracy maybe is a more important metric？Otherwise a method which only focuses on improving the accuracy on the memory would perform best for preventing catastrophic forgetting.



Unclear sentences and typos, just to list a few:

i. This implies that ∆f is not a reason for moving away from stationary
points of f by catastrophic forgetting

Typos:

i. gradient descent based algorithm reach -> gradient descent based algorithm reaches

ii. an replay memory -> a replay memory

iii. but use limited -> but uses limited

iv. g_j is not defined in page 2.

v.  we has trained ->  we have trained



**Summary Of The Paper:**

In this paper, the authors analyses the convergence rate of episodic memory-based continual learning methods. The authors formulate the continual learning problem as a nonconvex finite-sum optimization problem. Based on the analysis, the authors propose an adaptive learning rate scheduling methods to adjust the learning rates based on the gradients computed in each iteration. The results on several benchmarks show that the proposed method can achieve better results  than the baselines.

**Summary Of The Review:**

In this paper, the authors conduct a theoretical analysis of the memory based lifelong learning algorithms which is an important problem. However, the writing of the paper needs improvement and  the experimental results are not convincing enough to support the theory.

After response:

The authors' responses partly address my concerns. But in the current form of the paper, it is not ready to be published. The authors should conduct more experiments to make the theory more convincing.

---

> ### Author Response · Authors · 2021-11-23
> **Response to Reviewer UMAT**
>
> We appreciate your valuable review to improve our paper.
> We will revise the paper with clear statements, and correct typos that you comment.
> We have added the detailed discussion of the points that you have concerned in the updated revision by highlighting in red.
> The followings are a short summary of our additional comments on your mentions.
>
> > NCCL + reservoir (ring) underperforms ER-reservoir(ring). need discussion
>
> We note that our method has the smallest forgetting metric among all baselines.
> It implies that our method is focused on reducing catastrophic forgetting.
> We think that learning on the current task $C$ is not enough to show the best scores for the average accuracy.
> However, we emphasize that the results with larger memory outperforms the baelines.
> Then, we can conclude that our method suffers from the transfer effect with small memory.
> Because, the small subset cannot cover the gradients for entire datapoints.
>
> > prevent catastrophic forgetting, but average acc maybe is a more important metric?
>
> We consider that both average acc and the forgetting metric matters.
> By considering only average acc, we can underestimate the forgetting by reaching the high performance on the current task.
> Even if our method shows slightly lower value on average acc, we can observe that our algorithm shows the better perfomance on the forgetting metric.
>
>
> > $\Delta f$ is not a reason for moving away from stationary points of $f$ by catastrophic forgetting.
>
> We have added more explanation in the page 3 with red color.

---

> > ### Author Response · Authors · 2021-12-07
> > **We additionally uploaded the additional experiments to address your concerns.**
> >
> > Thank you again for your comments.
> >
> > We addressed the importance of forgetting metric, which is one of the valuable metric to evaluate the performance of continual learning in the above official comments for all reviewers and chairs.
> > We know that our theory based schme, NCCL suffers from learning new tasks slightly when the size of replay memory is tiny.
> > However, we want to emphasize that our method shows the best performance in terms of forgetting metric, which demonstates the amount of preventing forgetting.
> > This implies that our proposed optimization problem that minimizes the cumulative sum of catastrophic forgetting is the key to reduce the forgetting as NCCL learns new tasks simulateneously and reasonably.
> >
> > We partially agree that the notation is confusing, but this notation is originally form the literature of nonconvex optimization.
> >
> > If you have any ambiguites, concerns, and questions, please let us know.

---

### Official Review · Reviewer_jqxZ · 2021-11-02

**Correctness:** 3
**Technical Novelty And Significance:** 2
**Empirical Novelty And Significance:** 2
**Recommendation:** 5
**Confidence:** 3

**Main Review:**


Strengths

The use of adaptive learning rates in SGD provides a novel formulation on the continual learning problem. Experimental results show the merits of the proposes approach as compared to other state-the-art algorithms while a theoretical analysis of convergence is presented.

Remarks/Weaknesses
- The authors should make more clear the contributions of the current paper as compared to other relevant state-of-the-art algorithms possibly enumerating them.

- The writing in poor in some parts of the paper making it difficult for the reader to grasp the meaning.
E.g.
       a) last 3 sentences  of page 2 . What the authors mean by saying "..smaller than the loss gap of general SGD..."
       b) Section 3, First paragraph : "We use the convergence rate..., which denotes the IFO complexity ..."
- The two figs in Figure 2 are not well explained in the caption provided.

- The overfitting and catastrophic forgetting terms $B_t, \Gamma_t$ appears rather abruptly in the text and are not well explained.

- Subsection 3.3. First sentence:  "...the model $x^t $eventually converges to $M\cup C$, not $P\cup C$. It is not clear what the authors mean by saying that $x^t$ converges to $M\cup C$.

- Lemma 3: What does it mean that the convergence rate is ON $M\cup C$? This looks rather weird, again it is not clear at all what the author mean by that.






**Summary Of The Paper:**

The authors propose a stochastic gradient descent algorithm with adaptive learning rates for continual learning providing an  analysis of convergence. The main idea is to formulate the problem as a finite sum of non-convex objective functions with each component of the sum corresponding to a different task. The authors claim that the use of adaptive learning rates allow them to better control the relative importance of different tasks.

**Summary Of The Review:**

The paper proposes a  stochastic gradient descent algorithms applied on finite sums of nonconvex functions. The main novelty of the paper lies in the application of  the SGD ideas in the context of the continual learning.  In principle, this seems to be an interesting direction. However the  writing is rather poor in general and the authors fail to give an intuition of theoretical results presented in the paper. There also seems to be some wordings that make it difficult to understand some technical details of the paper (see above). I believe that the paper in its current form does not meet the high standards of ICLR.

--------------------------------------------
Comment after authors' responses:
I appreciate the efforts of the reviewers to address my comments and the changes they have made. However, I will keep my score to 5 since I believe the paper still lacks clarity of presentation and the significance of the contributions is not sufficiently illustrated. Moreover, the writing needs improvement and I encourage the authors to further work on it and resubmit an updated version of it to a future conference.

---

> ### Author Response · Authors · 2021-11-23
> **Response to Reviewer jqxZ**
>
> Thank you for the thoughtful comments on our paper.
> We have revised the paper for clarity on the points mentioned and added additional discussion of comparision with other relevant algorithms.
> We also address each point in greater detail below.
> Please let us know if our comments are not enough to address your concerns.
>
>
> > The authors should make more clear the contributions of the current paper as compared to other relevant state-of-the-art algorithms possibly enumerating them.
>
> We have added a discussion of the comparison with other baselines in page 9.
> We note that our main contribution is the theoretical convergence analysis of catastrophic forgetting (interference) and transfer.
> Previous works intuitively uses the concept of the inner product of gradients without any theoretical guarantee.
> Our analysis extends the conventional scheme of reducing the effect of interference (the negative value of inner product) to the optimization problem that benefit from increased transfer when $\langle \nabla f_{I_t} (x^t), \nabla g_{J_t}(x^t) \rangle > 0 $ and gradient cancelltation when $\langle \nabla f_{I_t} (x^t), \nabla g_{J_t}(x^t) \rangle < 0 $ by adaptive learning rates to reduce the cumulative sum of catastrophic forgetting term $\sum \Gamma_t $.
>
> > The writing is poor in some parts, which make readers hard to grasp the meaning.
>
> This was sloppy phrasing on our part.
> We have revised the following sentences and highlighted the polihsed sentences in red color.
>
>
> a) last 3 sentences of page 2 . What the authors mean by saying "..smaller than the loss g ap of general SGD..."
>
> Without the continual learning scenario, a general nonconvex SGD updates the parameters from an randomly initialized point, which is highly likely to have the large loss $f(x^0)$.
> Then, $\Delta_f$ is the key constant to determine the IFO complexity for converengece as $\Delta_f$ is in the numerator of Equation 11.
> However, a continual learning algorithm has already converged to a local optimal point $x^0$ for the previous task $f(x)$ and might get a much smaller $\Delta_f$ than the general SGD.
> It means that $\Delta_f$ for nonconvex continual learining in Equation 11 dose not have a large impact on the IFO complexity.
> To generalize the theoretical result, we deine the worst local minimum to explain the upper bound of convergence rate.
>
> b) Section 3, First paragraph : "We use the convergence rate..., which denotes the IFO comp lexity ..."
>
> We add a citation at the end of the sentence to introduce the concept of the relation between IFO complexity and convergence rate.
>
> > The two figs in Figure 2 anre not well explained.
>
> We think that you mentioned about Figure 1.
> We agree that this figure is weak to explain the concept of our method.
> Then, we moved the figure to appnedix B.
> The orginial intention to show the figure is that it is hard to recover the stationary if the continual learning algorithm encounters an interference.
> The right figure explains that we can reduce the interence and boost the transfer by using adaptive learning rates.
>
>
> > $B_t$ and $\Gamma_t$ appears rather abuptly and not well explained
>
> We added some explnation below the Equation 8 for readibility.
>
> > What is the meaning that $x_t$ converges to $M \cup C$.
>
> With the given memory $M$, we want to emphasize that the overfitting on $M$ occurs for the inifite iterations.
> However, it was a sloppy phrase.
> We remove this sentence in the revised version.

---

### Author Response · Authors · 2021-12-01
**Addition experimental results on the quantitative analysis of catastrophic forgetting**

First of all, we would like to thank you for your patience.

We provide the following additional experimental results, which Reviewer TD2U suggests, to help reviewers to quantitively compare the cumulative sum of catastrophic forgetting term $\sum_{t}E_{I_t}E_{J_t}[\Gamma_t|M_0]]$.
All experiments conduct a single-pass over the data stream.
Batch size of data stream and replay memory are both 10.

## Results

**Table 1: Multi-headed split-CIFAR100, reduced size Resnet-18 $n_f=20$. The cumulative sums of $E_{I_t}E_{J_t}[\Gamma_t|M_0]]$ for a given $M_0$ at the end of each epoch**

|       Method       |  $\sum_{t} E_{I_t}E_{J_t}[\Gamma_t]]$  |          |          |          |          |          |          |          |          |          |          |          |          |          |          |          |          |          |          |          |   accuracy  |  forgetting |
|:------------------:|:---------------------:|:--------:|:--------:|:--------:|:--------:|:--------:|:--------:|:--------:|:--------:|:--------:|:--------:|:--------:|:--------:|:--------:|:--------:|:--------:|:--------:|:--------:|:--------:|:--------:|:-----------:|:-----------:|
|       epochs       |           1           |     2    |     3    |     4    |     5    |     6    |     7    |     8    |     9    |    10    |    11    |    12    |    13    |    14    |    15    |    16    |    17    |    18    |    19    |    20    |             |             |
|        A-GEM       |           0           | 454.3601 | 1032.534 | 1565.519 | 2324.911 | 2957.218 | 3652.587 | 4318.981 |  4923.44 | 5588.325 | 6123.903 | 6804.472 | 7303.999 | 7987.095 | 8555.443 |  9177.25 | 9884.811 | 10536.82 |  11098.9 | 11697.92 |  50.7(2.32) |  0.19(0.04) |
|       ER-Ring      |           0           | 8.813561 | 17.43859 | 27.04981 | 35.46741 | 44.34798 | 52.90565 | 64.31162 | 71.72681 | 83.29379 | 94.36832 | 107.6595 | 116.8971 | 126.5865 | 137.1054 | 149.0067 | 163.2852 | 172.8176 | 183.8802 | 194.2378 |  56.2(1.93) |  0.13(0.01) |
|    ER-Reservoir    |           0           | 402.7679 | 1331.371 | 2598.568 | 4016.047 | 5443.711 | 6624.204 | 7927.677 | 9097.198 | 10388.36 | 11588.99 | 12714.19 | 13642.11 |  14734.8 | 15815.43 | 16812.42 | 17956.68 | 18936.14 | 20202.46 | 21101.88 |  46.9(0.76) |  0.21(0.03) |
|     NCCL + Ring    |           0           | 4.151815 | 8.369048 |  15.0003 | 22.41685 | 7.796947 | -0.37874 |  5.4692  | 7.210486 | 16.13309 | 22.55482 | 30.99277 |  38.1405 | 45.13828 | 53.81658 | 60.18736 | 69.07559 | 79.78552 | 71.44843 | **78.33006** | 54.63(0.65) | **0.059(0.01)** |
| NCCL +   Reservoir |           0           | -77.2628 | -42.7353 | -46.1527 | -14.6228 | 21.73526 | 27.72568 | 57.72162 | 91.32076 | 129.0324 | 153.4561 | 184.2096 | 213.2439 | 245.6522 | 274.3416 | 308.2475 | 344.7879 | 380.7963 | 415.2257 | 442.5131 | 52.18(0.48) | 0.118(0.01) |


## Discussions

We first note that NCCL variants show the best performance on the forggeting metric, and NCCL-Ring shows the lowest cumulative sum of $\Gamma_t$.
It means that the forgetting is successfully suprressed by our adative learning rate schemes with the proposed nonconvex optimization problem in Equation (14) of our paper.
We also note that the cumulative sum is also highly correlated with both the average test accuracy and the forgetting metric.
It is remarkable that NCCL + Ring shows the lowest value when we compared to A-GEM, ER-Ring.
We also conclude that NCCL + Ring slightly suffers from learning new tasks by suprressing their gradient when the transfer occurs $\langle \nabla f_{I_t}(x^t), \nabla g_{J_t}(x^t) \rangle>0$ when we comapres to vanilla ER-Ring.

---

> ### Author Response · Authors · 2021-12-01
> **Additional results on the overfitting to memory**
>
> **Table 2: (NCCL-Ring) Multi-headed split-CIFAR100, reduced size Resnet-18 $n_f=20$. The estimated mean and std of $B_t$ over 10 random choices of $M_0$ at the end of each epoch**
>
> | epochs | $E[B_t]$ |          |          |          |          |          |          |          |          |          |          |          |          |          |          |          |          |          |          |          |
> |--------|----------|----------|----------|----------|----------|----------|----------|----------|----------|----------|----------|----------|----------|----------|----------|----------|----------|----------|----------|----------|
> |        | 1        | 2        | 3        | 4        | 5        | 6        | 7        | 8        | 9        | 10       | 11       | 12       | 13       | 14       | 15       | 16       | 17       | 18       | 19       | 20       |
> | mean   | 0        | -0.14675 | -0.45504 | -0.55479 | -0.29252 | -0.34124 | -0.47483 | -1.17503 | -0.34057 | -0.94361 | -0.37363 | -0.32855 | -0.14306 | -0.30286 | -0.47138 | -0.54014 | -0.47511 | -0.502   | -0.79833 | -0.43741 |
> | std    | 0        | 0.075992 | 1.062644 | 0.811357 | 0.181361 | 0.361017 | 0.412218 | 1.101363 | 0.27188  | 1.350875 | 0.27878  | 0.232343 | 0.09332  | 0.257546 | 0.29631  | 0.329638 | 0.560819 | 0.372894 | 0.84614  | 0.176997 |
>
> Table 2 shows that the estimated values of mean and standard deviation over 10 random chocies of $M_0$ with different seeds.
> By our theoretical result in Algorithm 1, values at each epoch are computed by all previously learned tasks $P=\cup_{k} D_k$ that the continual learning agent has seen for $k$ tasks, respectively.
> We note that the estimated means are negative values which are very close to zero.
> It implies that the empirical result of Lemma 1, which shows the effect of $B_t$ on Equation (10) vanishes by $E[B_t]=0$, demonstrates that the theoretical analysis on the expectation over the random choices of $M_0$ is correct.
> By considering the distribution, some of trials degrades the convergence by increasing the upper bound with the positive overfitting term, but the other trials seems to help the continual learning agent to converge on $f(x)$.
> We conclude that the overfitting bias is a minor factor of degrading the performance of continual learning agent when we compare to the catastrophic forgetting term $\Gamma_t$, and $B_t$ itself slightly helps to keep the convergence on $f$ empricially.
> The forgetting process forces the agent to be far from the stationary point on $f(x)$, then the performance degradation by overfitting on the specific datapoints in the replay memory might be hard to occur in terms of convergence analysis.

---

### Decision · Program_Chairs · 2022-01-20

**Decision:**

Reject

**Comment:**

This paper theoretically studies the convergence of memory-based continual learning with stochastic gradient descent, and suggested several methods based on adaptive learning rates.

The reviewers appreciated the novelty of the direction, and some of them thought the experimental results are promising.

However, most reviewers (3/4) were negative. I think the main reason was the paper presentation and clarity, which they found lacking (and I agree). One reviewer thought the experimental evaluation should be improved, but there might have been some misunderstanding there. Lastly, even the positive reviewer thought the results were somewhat incremental and non-surprising.

I hope the authors improve their paper and re-submit.